# UQ-Merge: Uncertainty Guided Multimodal Large Language Model Merging

## Abstract

Multimodal Large Language Models (MLLMs) have gained increasing popularity as a promising framework for leveraging the strong language reasoning capabilities in the vision-language domain. Given a wide range of MLLMs, model merging potentially offers a cheap way to aggregate their diverse knowledge into a single MLLM. However, directly plug-in existing model merging approaches often leads to suboptimal performance due to (1) inclusion of harmful models that have over-confident predictions in the target task; (2) the lack of specialized designs for vision-language inputs. To tackle these pain points, we conduct pioneering investigations to dissect the merging procedures and propose an uncertainty-guided MLLM merging algorithm, *i.e.*, UQ-Merge, which $i$) identifies beneficial candidates for merging, $ii$) determines the merging order and the number of helpful candidates, and $iii$) performs appropriate merging. Within our framework, we consider uncertainty quantification on both text and vision inputs to examine the MLLM prediction confidence, and then decide whether and when a MLLM needs to be included. It is worth mentioning that our vision-language uncertainty quantification does not require access to sample labels, making it more practical in various scenarios. Extensive experiments consistently demonstrate the superior MLLM merging performance of UQ-Merge in both held-in and held-out vision-language benchmarks. For example, compared to existing state-of-the-art merging methods, UQ-Merge brings substantial performance improvements of up to 44.3% on average accuracy in 12 datasets. Codes are available at https://anonymous.4open.science/r/UQ-Merge-7CD7.

## 1 Introduction

Multimodal Large Language Models (MLLMs) have achieved numerous successes in various visual-language tasks including visual reasoning (Yin et al., 2023), autonomous driving (Cui et al., 2024), visual question answering (Zhang et al., 2024a), *etc*. A popular paradigm to reach impressive vision-language reasoning capabilities typically combines a LLM backbone with a pretrained vision encoder (Alayrac et al., 2022; Liu et al., 2024b;a; McKinzie et al., 2024; Tong et al., 2024; Xue et al., 2024). Fine-tuning of pre-trained MLLMs has been explored in many vision-language domains like biomedicine answering (Li et al., 2024b) and text-rich image understanding (Zhang et al., 2023), pushing the need to incorporate knowledge from diverse domains. To achieve this, rather than collecting all datasets and spending massive computing costs to train a new model from scratch, model merging has been widely explored as a method to overcome high training costs and aggregate knowledge from different datasets, by leveraging existing models and merging them in a training-free manner. Existing studies have shown superior merg-

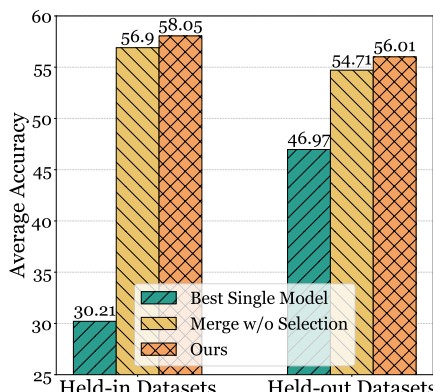

Figure 1: The average accuracy of the best single model, merge all models, and merge UQ-Merge selected models. Held-in datasets refer to datasets used for UQ. Held-out datasets are benchmarks unused for model selection.

ing results across tasks, highlighting its advantage of efficiently integrating separate advancements (Ding et al., 2024; Goddard et al., 2024; Wan et al., 2024; Yang et al., 2024; Lu et al., 2024).

However, the model merging in the MLLM domain remains less explored. To begin with, we apply a single-modal merging method (Figure 1 (yellow)) and it achieves stronger performance compared to the best single model (Figure 1 (green)). Meanwhile, we observe that by selectively merging MLLMs (Figure 1 (orange)), the performance of the merged model can be further improved.

Despite the performance gain, applying single-modal merging methods on MLLM merging has limitations. **Firstly**, existing merging methods assume that all models are beneficial for merging performance. As pointed out by (Zhao et al., 2024) and observed in Figure 1, this assumption may not hold true in real-world scenarios where models to merge are trained on divergent datasets. Some models produce over-confident predictions on target tasks and merging them will result in a performance decrease of the merged model. **Secondly**, these merging methods are designed solely to focus on single-modal model merging. Given these limitations, an ideal MLLM merge mechanism should be selective and aware of multimodal inputs.

To address these challenges, we propose `UQ-Merge`, an uncertainty quantification guided MLLM merging algorithm that features vision-language optimized design to ameliorate performance degradation caused by merging over-confident models. Specifically, `UQ-Merge` ❶ uses image-text perturbation-based uncertainty quantification (UQ) to evaluate models, and ❷ sorts models by the descending order of uncertainty to reduce the impact of over-confident models. ❸ `UQ-Merge` incrementally enlarge the group of models to merge, and ❹ return the merged model with the lowest uncertainty. Our contributions are summarized as follows:

- Due to the inclusion of over-confident models and the lack of vision-language specific designs, directly applying single-modal merging methods results in suboptimal performance. To resolve these issues, we conduct pioneering work in the MLLM field.

- To investigate design factors influencing MLLM merging performance, we raise and answer research questions: What is a more effective metric for selecting helpful models? How to decide the merging order and select models? How to implement UQ for MLLM? And how to appropriately merge selected models?

- We propose an MLLM-tailored image-text perturbation-based uncertainty quantification method and introduce `UQ-Merge`, an uncertainty guided MLLM merging method that could identify and exclude over-confident models.

- Experiments demonstrate that `UQ-Merge` consistently outperforms single-modal merging methods. With the same number of models used for merging, `UQ-Merge` achieves an average accuracy improvement of 2.62% on held-in datasets and 1.06% on held-out datasets compared to existing merging methods. Furthermore, `UQ-Merge` can surpass single-modal merging methods that have access to more models, by 0.54% on held-in datasets and 1.3% on held-out datasets.

## 2 RELATED WORK

**Multimodal Large Language Models (MLLMs).** Large Language Models (LLMs) have demonstrated strong reasoning and instruction-following capability (Zhao et al., 2023; Touvron et al., 2023a;b). In light of this, many works (Alayrac et al., 2022; Liu et al., 2024b;a; McKinzie et al., 2024; Tong et al., 2024; Xue et al., 2024) propose to further incorporate pre-trained vision backbones (Radford et al., 2021; Zhai et al., 2023) to enable visual perception capabilities in existing LLMs, producing Multimodal Large Language Models (MLLMs). These models extend the powerful capabilities of LLMs into the domain of visual comprehension and reasoning. The mainstream architecture of MLLMs generally consists of three components: a vision encoder that extracts features from visual inputs, a modality adapter that projects the outputs of the vision encoder into the token embedding space of the LLM backbone, and an LLM backbone that processes both image and text inputs to generate responses (Yin et al., 2023; Zhang et al., 2024a). Modality adapter implementations include projection-based, query-based, and fusion-based variants (Zhao et al., 2023; Li et al., 2023b; Radford et al., 2021; Alayrac et al., 2022). The typical training process of an MLLM involves two stages: pre-training and instruction tuning. During the pre-training stage, the vision encoder and the LLM are kept frozen, while the adapter is trained on a large corpus of image-text

pairs. The objective of this stage is to train the adapter so that visual tokens can be effectively embedded into the language token space. Following pre-training, visual instruction tuning (Liu et al., 2024b;a) is conducted using instruction-following examples from diverse vision question answering (VQA) tasks. This step aims to improve the model's ability to follow instructions in VQA or image captioning scenarios. Given individual MLLMs, it remains under-explored how to leverage these mdoels and aggregate their knowledge. Motivated by this, we propose `UQ-Merge`, a UQ-based MLLM merging method to incorporate models fine-tuned on different tasks.

**Model Merging.** Model Merging (Ainsworth et al., 2023) combines multiple pre-trained or fine-tuned models into a unified, powerful model, leveraging the strengths of specialized models while maintaining versatility without requiring additional training. Early zero-shot merging methods, such as weight averaging and Linear Mode Connectivity (Nagarajan & Kolter, 2021; Wortsman et al., 2022), laid the foundation for this approach. Task Arithmetic Ilharco et al. (2023) manipulates task vectors for effective merging, while TIES (Yadav et al., 2023) addresses parameter interference through trimming and conflict resolution. DARE (Yu et al., 2024) selectively optimizes parameters to enhance merging without extra training, utilizing the geometric properties of weights (Shoemake, 1985; Jang et al., 2024). In the latest works, DELLA merges models by pruning and re-scaling weights based on their magnitude (Deep et al., 2024), and Model Stock finds the optimal interpolation ratio between merging candidates, using a pre-trained model to identify a robust anchor point (Jang et al., 2024). In the multimodal domain, model merging has similarly proven its ability to transform modality-specific models into modality-agnostic models (Sung et al., 2023). These existing merging studies motivate us to explore model merging in the MLLM domain.

**Uncertainty Quantification (UQ).** Uncertainty quantification (UQ) in predictions from deep neural networks (DNNs) has been a longstanding and essential problem (Abdar et al., 2021; Gawlikowski et al., 2023). The sources of uncertainty can be categorized into data uncertainty (aleatoric uncertainty) and model uncertainty (epistemic uncertainty). Broadly, UQ methods can be categorized into four groups (Gawlikowski et al., 2023): single-inference deterministic methods (Nandy et al., 2020; Oala et al., 2020; Sensoy et al., 2018), Bayesian neural network (BNN) methods (Gal & Ghahramani, 2016; Loquercio et al., 2020), ensemble-based methods (Rahaman et al., 2021; Lakshminarayanan et al., 2017) and test-time augmentation methods (Ayhan & Berens, 2018; Ashukha et al., 2020). For UQ in LLMs, Sampling with Perturbation for UQ (SPUQ) (Gao et al., 2024) is a test-time augmentation method that generates a set of perturbed prompts and quantifies uncertainty based on the similarity between the responses. In the MLLM domain UQ is less explored. One recent work applies conformal prediction (CP) for UQ in MLLMs (Ye et al., 2024; Kostumov et al., 2024). However, the CP method requires labeled data to estimate the model's uncertainty, which is infeasible in many real-world applications due to the lack of ground truth. In (Daheim et al., 2023), the authors propose to utilize gradient-based UQ to mitigate mismatches of gradients when merging models trained on various tasks. However, it still requires labels to compute the gradients and needed Hessian matrices To address this, we propose a vision-language perturbation-based UQ method for MLLM that does not require labels.

# 3 METHODOLOGY

## 3.1 PRELIMINARIES

**The Architecture Overview of Multimodal Large Language Model.** The definition of Multimodal Large Language Models (MLLMs) is LLM-based models with the ability to receive, reason, and output with multimodal information (Yin et al., 2023). Prior to MLLMs, many works were devoted to multimodality learning (Radford et al., 2021; Li et al., 2021; Wang et al., 2021). In this paper, we focus on MLLMs that process image-text inputs and use $(x_v, x_t)$ to represent an input image $x_v$ and text $x_t$ pair to an MLLM $\mathcal{M}(\cdot, \cdot)$. The most common MLLM architecture for image-text inputs (Liu et al., 2024b;a; Chen et al., 2024; McKinzie et al., 2024; Tong et al., 2024) typically comprises a pre-trained vision encoder $\mathcal{V}(\cdot)$, an adapter $\mathcal{A}(\cdot)$ and an LLM backbone $\mathcal{F}(\cdot)$. An overview of the model architecture is provided in Figure 2 (a). The text input $x_t$ is split into textual tokens $\boldsymbol{h}_t$. The vision encoder extracts visual features from the input image $x_v$, represented as visual tokens $\boldsymbol{z}_v = \mathcal{V}(x_v)$, which are then mapped by the adapter into the embedding space of language tokens, yielding $\boldsymbol{h}_v = \mathcal{A}(\boldsymbol{z}_v)$. The LLM processes both visual tokens $\boldsymbol{h}_v$ and language tokens $\boldsymbol{h}_t$ to generate an output $\mathcal{F}(\boldsymbol{h}_v, \boldsymbol{h}_t)$ to the textual query.

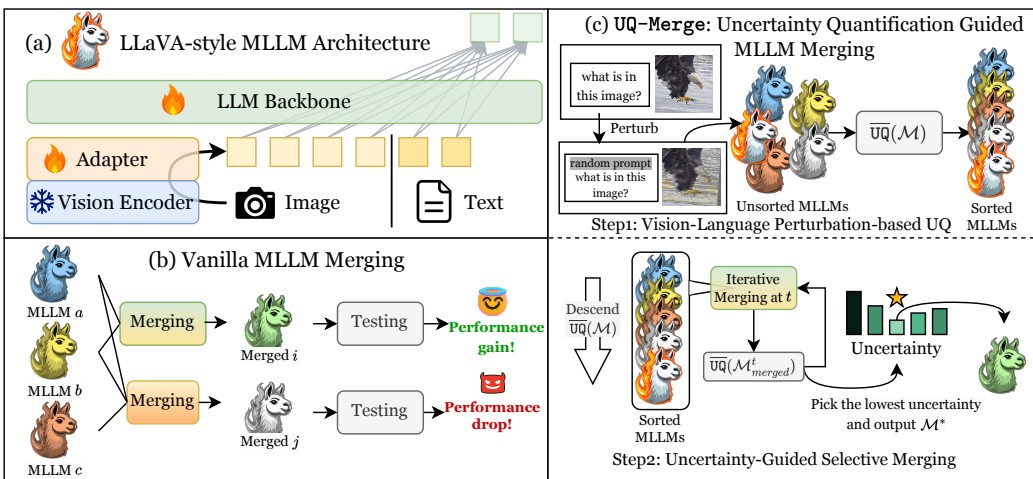

Figure 2: The overview of our proposed `UQ-Merge`. (a) illustrates the common architecture of MLLMs. In (b) it shows vanilla merging fails when no selection is considered, and in contrast performance gains by selectively merging. (c) shows the two steps in `UQ-Merge`: (1) `UQ-Merge` quantifies and sorts models by descending order of their uncertainty. (2) After sorting, each model is gradually included and merged, and returns the model with the lowest uncertainty.

**Model Merging.** The goal of model merging is to aggregate knowledge from two or more models with the same architecture into a unified model that retains the strengths and capabilities of the original models. Formally, given a set of models $\{\mathcal{M}_1, \ldots, \mathcal{M}_n\}$, model merging can be expressed as $\mathcal{M}^* = \text{Merge}(\mathcal{M}_1, \ldots, \mathcal{M}_n)$, where $\mathcal{M}^*$ represents the merged model and $\text{Merge}(\cdot)$ is a merging method. In MLLM merging, as the vision encoders $\mathcal{V}$ of models with the same architecture are usually initialized from the same pre-trained model and kept frozen during the pre-training and fine-tuning process (Liu et al., 2024b;a; Lin et al., 2024; Xue et al., 2024), their weights are identical and do not require merging. For this reason, $\text{Merge}(\cdot)$ only considers the adapter $\mathcal{A}$ and the LLM backbone $\mathcal{F}$ when applied on MLLMs.

### 3.2 UQ-MERGE: UNCERTAINTY QUANTIFICATION (UQ) GUIDED MLLM MERGING

To overcome the aforementioned challenges of over-confident merging candidate models and the lack of vision-language oriented merging method, we propose `UQ-Merge`, which consists of a vision-language perturbation-based MLLM uncertainty quantification (Section 3.3) to evaluate models, and a merging algorithm based on the uncertainty of models (Section 3.4). The procedure of `UQ-Merge` is described in Figure 2 (c). First, UQ is applied to MLLMs to quantify their uncertainty, and models are sorted in descending order of uncertainty to later consider potentially over-confident models. Then, `UQ-Merge` incrementally merge the sorted models and record the uncertainty scores of the merged model at each step. Finally, the merged model with the lowest uncertainty during the process is selected as the final output. Throughout the process, `UQ-Merge` adopts the same UQ function described in Section 3.3.

### 3.3 VISION-LANGUAGE PERTURBATION-BASED UNCERTAINTY QUANTIFICATION

Uncertainty quantification (UQ) (Mehrtash et al., 2020; Guo et al., 2024; Gao et al., 2024) has demonstrated superior effectiveness in evaluating models without labels, which is highly practical in real-world scenarios. UQ provides a quantified score for a model, indicating its confidence level reliability and performance (Wang et al., 2022; Si et al., 2023). In light of this, we develop a vision-language perturbation-based UQ to evaluate MLLMs for model merging. Specifically, given input image-text pair $x_v$ and $x_t$, our perturbation-based MLLM $\text{UQ}(\cdot, \cdot, \cdot)$ on a MLLM model $\mathcal{M}$ of this sample is defined as:

$$
\underbrace{\text{UQ}(\mathcal{M}, x_v, x_t)}_{\text{Model uncertainty}} \approx \underbrace{\mathcal{H}\left(\frac{1}{J}\sum_{j=1}^{J}\mathcal{M}_\epsilon^j\left(\mathcal{P}_v^j(x_v), \mathcal{P}_t^j(x_t)\right)\right)}_{\text{Total uncertainty}} - \underbrace{\frac{1}{J}\sum_{j=1}^{J}\mathcal{H}\left(\mathcal{M}_\epsilon^j\left(\mathcal{P}_v^j(x_v), \mathcal{P}_t^j(x_t)\right)\right)}_{\text{Data uncertainty}},
$$

(1)

where $J$ is the number of perturbations, $x_v$ and $x_t$ are original image-text input, $\mathcal{M}_\epsilon^j$ is the $j_{\text{th}}$ perturbed model, $\mathcal{P}_v^j(\cdot)$ is the $j_{\text{th}}$ perturbation function for the image input, $\mathcal{P}_t^j(\cdot)$ is the $j_{\text{th}}$ text perturbation function, and $\mathcal{H}(\cdot)$ is the entropy function. The perturbed model $\mathcal{M}_\epsilon^j$ is derived by adding Dropout (Srivastava et al., 2014) to the attention score. We implement $\mathcal{P}_v^j(\cdot)$ as a composition of image transformation functions such as Shear, Translate, Rotate, Equalize, and Posterize (Cubuk et al., 2018; Hendrycks et al., 2020). $\mathcal{P}_t^j(\cdot)$ is implemented by adding randomly selected prompts (e.g., "you are a helpful assistant") to the original text input. Following previous works (Ye et al., 2024; Kostumov et al., 2024), we employ the prompt[1] to ask the model to answer directly with an option and extract the logits of option letters from the first newly generated token, and entropy is computed on the logits. The model uncertainty is the difference between total and data uncertainty, where the total uncertainty is the entropy of the average prediction, and the data uncertainty is the average entropy of each prediction. In the literature below, we use $\overline{\text{UQ}}(\mathcal{M})$ to represent the average uncertainty of $\mathcal{M}$ over samples by using $\text{UQ}(\cdot, \cdot, \cdot)$.

### 3.4 Uncertainty-Guided Merging for Model Selection

Given models to merge, we sort the models in descending order of uncertainty to reduce the impact of over-confident models for merging (**Step (1)** in Figure 2 (c)). Starting from the model with the lowest uncertainty, `UQ-Merge` gradually considers each model. At each step, one model $\mathcal{M}_i$ is added to the merging group, and merging method $\text{Merge}(\cdot)$ is employed to produce a merged model $\mathcal{M}_{\text{merged}}$, and $\overline{\text{UQ}}(\cdot)$ is applied to quantify its uncertainty. In our practice of `UQ-Merge`, $\text{Merge}(\cdot)$ is implemented as linearly averaging the weights of all models. `UQ-Merge` allows different choices of merging functions, but as will be shown in Table 6, linear merging is simple and brings strong

---

**Algorithm 1** `UQ-Merge`

1: **Input:** Models $\{\mathcal{M}_1, \ldots, \mathcal{M}_n\}$, UQ function $\overline{\text{UQ}}(\cdot)$,
         Merging method $\text{Merge}(\cdot)$
2: **Output:** Merged model $\mathcal{M}^*$
3: Compute vision-language uncertainty $\{u_1, \cdots, u_n\}$ for
   each model in $\{\mathcal{M}_1, \ldots, \mathcal{M}_n\}$
4: $\{\mathcal{M}_1', \ldots, \mathcal{M}_n'\} \leftarrow$ Sort $\{\mathcal{M}_1, \ldots, \mathcal{M}_n\}$ by descending
   order of vision-language uncertainty $\{u_1, \cdots, u_n\}$
5: Initialize the uncertainty of the merged model as $u^* \leftarrow \infty$
6: Initialize the merged model as $\mathcal{M}^* \leftarrow \mathcal{M}_1'$
7: **for** a model $\mathcal{M}_t'$ in $\{\mathcal{M}_1', \ldots, \mathcal{M}_n'\}$ **do**
8:     $\mathcal{M}_{\text{merged}}^t \leftarrow \text{Merge}(\mathcal{M}_1', \ldots, \mathcal{M}_t')$
9:     $u_{\text{merged}}^t \leftarrow \overline{\text{UQ}}(\mathcal{M}_{\text{merged}}^t)$
10:     **if** $u_{\text{merged}}^t < u^*$ **then**
11:         $\mathcal{M}^* \leftarrow \mathcal{M}_{\text{merged}}^t; u^* \leftarrow u_{\text{merged}}^t$
12:     **end if**
13: **end for**
14: **return** $\mathcal{M}^*$

---

performance. After using all models, `UQ-Merge` returns the merged model with the lowest uncertainty (**Step (2)** in Figure 2 (c)). As the merged model aggregates knowledge from diverse domains, we view low uncertainty after merging as a signal of strong capability on tasks and select the model.

## 4 Experiments

### 4.1 Implementation Details

**Model Preparation.** In our experiments, we begin with the pre-trained LLaVA-v1.5-7B model (Liu et al., 2024a), which utilizes Vicuna-1.5-7B (Chiang et al., 2023) as the LLM backbone $\mathcal{F}$, a CLIP-ViT-L-336px (Radford et al., 2021) as the vision encoder $\mathcal{V}$, and a two-layer MLP with a hidden dimension of 4096 as the modality adapter $\mathcal{A}$. Then the pre-trained model is fine-tuned with instruction-tuning datasets that focus on diverse vision-language capabilities to create the models for merging. Each model is trained on a distinct dataset. Specifically, we follow the instruction-tuning practices of LLaVA-v1.5 and use the same datasets, which can be categorized into: visual reasoning datasets (Hudson & Manning, 2019; Kazemzadeh et al., 2014; Mao et al., 2016); text-rich datasets (Mishra et al., 2019; Sidorov et al., 2020); knowledge-based VQA datasets (Marino et al., 2019; Schwenk et al., 2022); GPT-generated datasets (Liu et al., 2024b; ShareGPT, 2023); and general VQA datasets (Goyal et al., 2017; Krishna et al., 2017). All models are trained following the default training configuration from LLaVA-v1.5-7B, using AdamW (Loshchilov & Hutter, 2019)

---

[1]"Answer with the option's letter from the given choices directly."

as the optimizer and the learning rate starts from $2 \times 10^{-5}$ and decreases according to a cosine annealing scheduler. Models are trained in the distribution of $4 \times$ A100 GPUs using DeepSpeed ZeRO-3 (Aminabadi et al., 2022) with gradient checkpointing enabled, and the batch size per device is set to 16. On each fine-tuning dataset, the pre-trained model is fine-tuned for 1 epoch.

**Single-Modal Baselines.** For sufficient comparison with our method that uses UQ to guide MLLM merging, we compare `UQ-Merge` against various single-modal merging methods and test their performance in multimodal scenarios. Specifically, we consider DARE (Yu et al., 2024), DELLA (Deep et al., 2024), Linear (Wortsman et al., 2022), TIES (Yadav et al., 2023), Task Arithmetic (Ilharco et al., 2023), and Model Stock (Jang et al., 2024). Due to the lack of model selection capability, we compare these methods in ❶ average performance in random selections and ❷ merge all models. Although baseline methods were originally designed for single-modal merging, they are capable of merging models that have the same architecture. We consider adapter and LLM backbone when using baselines, as a naive extension of these methods.

**Vision-Language Classification Datasets for Uncertainty Quantification.** We select vision-language classification datasets as our benchmarks for $\overline{\text{UQ}}(\cdot)$. Specifically, five datasets across five domains are considered: MMBench (reasoning / perception (Liu et al., 2023a)), OODCV-VQA (out-of-distribution robustness (Zhao et al., 2022)), ScienceQA (world knowledge (Lu et al., 2022)), SEEDBench (spatial and temporal understanding (Li et al., 2023a)), and AI2D (diagrams (Kembhavi et al., 2016)). In line with (Ye et al., 2024; Kostumov et al., 2024), we reformat the answers of these datasets and introduce two additional choices, "I don't know" and "None of the above," to the list of options. Since our vision-language perturbation-based UQ does not require labels, we treat these datasets as *held-in datasets* and also use them for the evaluation of merged models' performance in vision-language classification format tasks.

**Vision-Language Generation Datasets for Evaluation of Multimodal Capability.** To more comprehensively evaluate the merged models' performance, we choose seven vision-language generation tasks of six domains, including open real-world knowledge (OKVQA (Marino et al., 2019), MMMU (Yue et al., 2024)), text understanding (TextVQA (Singh et al., 2019)), compositional questioning answering (GQA (Hudson & Manning, 2019)), low-quality image understanding (VizWiz (Gurari et al., 2018)), general visual QA (VQAv2 (Goyal et al., 2017)), and hallucination (POPE (Li et al., 2023c)) as the benchmarks. As these datasets are not used for model selection in `UQ-Merge`, in the literature below we refer to these datasets as *held-out* datasets.

### 4.2 `UQ-MERGE` IS EFFECTIVE FOR REMOVING HARMFUL MODELS

In this section, we compare our `UQ-Merge` against various single-modal merging methods on held-in and held-out datasets to show the effectiveness of `UQ-Merge` in excluding harmful models. For baseline merging methods, we evaluate them by measuring the average performance of their merged models. For all baseline methods, each time the merged model is produced by merging a random model selection from all models we fine-tuned, and the number of models selected each time is the same as the selection of our method `UQ-Merge`. Evaluation results are summarized in Table 13 and Figure 3. From the results, the following observations can be drawn: ❶ Our `UQ-Merge` demonstrates superior performance compared to all other merging methods. Specifically, `UQ-Merge` achieves $2.62\% \sim 44.3\%$ and $1.06\% \sim 43.18\%$ improvement on average accuracy of held-in and held-out datasets. In fact, the performance of `UQ-Merge` even surpasses the maximum value among all baseline methods, as shown in Figure 3. This validates the effectiveness of `UQ-Merge` in model selection to exclude over-confident models. ❷ On the held-in dataset, which is used for uncertainty quantification, `UQ-Merge` obtains a more significant performance gain compared to held-out datasets, with 4 out of 5 highest accuracy. This justifies our practice of using UQ to perform model selection, as in real-world applications labels are usually unavailable and UQ only relies on input to evaluate a model, and UQ-guided model selection can effectively improve the performance on these applications and even generalize to held-out datasets. ❸ The performance of single-modal merging methods varies a lot in MLLM merging. The gap of average accuracy for baselines is $41.68\%$ and $42.12\%$ on held-in and held-out datasets, respectively. These large gaps show the various effectiveness of state-of-the-art single-modal merging methods when the setting is shifted to the MLLM merging.

Table 1: The comparison between `UQ-Merge`, single-modal merging methods and LLaVA-v1.5 that trains on the combined dataset. **Baseline methods merge randomly selected the same number of models to `UQ-Merge`.** Average and standard error of the accuracy of baselines across selections are reported. Results are measured with 3 selections. The best and second-best performances are highlighted in **bold** and underline, respectively.

| Merging Methods | Vision-Language Classification Datasets | | | | | |
| --- | --- | --- | --- | --- | --- | --- |
| | Average | AI2D | ScienceQA | SeedBench | MMBench | OOD-CV |
| DELLA | 13.75 ± 3.09 | 8.01 ± 6.94 | 2.72 ± 2.60 | 5.73 ± 4.96 | 26.25 ± 0.75 | 26.04 ± 0.48 |
| Linear | 53.15 ± 5.72 | 43.23 ± 7.63 | 57.97 ± 8.06 | 55.13 ± 7.55 | 70.26 ± 1.31 | 39.14 ± 4.95 |
| TIES | 55.43 ± 3.88 | 49.41 ± 3.64 | 64.15 ± 6.43 | 57.16 ± 5.07 | 69.82 ± 0.62 | 36.60 ± 3.70 |
| Task Arithmetic | 51.10 ± 7.23 | 43.58 ± 7.81 | 52.40 ± 6.88 | 48.58 ± 14.28 | 70.02 ± 1.39 | **40.95 ± 6.07** |
| Model Stock | 21.73 ± 7.27 | 18.22 ± 11.31 | 17.17 ± 19.61 | 20.35 ± 13.45 | 27.60 ± 1.14 | 25.32 ± 1.09 |
| | 49.99 ± 0.58 | 43.81 ± 0.92 | 65.07 ± 0.06 | 49.11 ± 0.84 | 64.10 ± 0.82 | 27.86 ± 0.52 |
| Ours | **58.05** | **51.75** | **68.07** | **60.56** | **70.35** | 39.52 |
| LLaVA-v1.5-7B | 64.96 | 54.79 | 70.43 | 60.49 | 72.04 | 67.05 |

| Merging Methods | Vision-Language Generation Datasets | | | | | | |
| --- | --- | --- | --- | --- | --- | --- | --- |
| | Average | OKVQA | TextVQA | GQA | MMMU | VizWiz | VQAv2 | POPE |
| DARE | 12.83 ± 8.01 | 0.37 ± 0.35 | 5.48 ± 4.76 | 10.91 ± 9.82 | 26.22 ± 0.80 | 2.43 ± 2.11 | 19.77 ± 17.35 | 24.60 ± 21.64 |
| DELLA | 47.94 ± 3.75 | 40.53 ± 8.05 | 41.17 ± 1.77 | 43.96 ± 3.26 | 32.78 ± 1.31 | 41.92 ± 5.66 | 65.33 ± 3.46 | 69.87 ± 18.26 |
| Linear | 54.13 ± 0.72 | 44.61 ± 5.90 | 42.74 ± 1.83 | 49.83 ± 0.92 | 33.71 ± 0.95 | 52.80 ± 4.44 | 69.52 ± 1.77 | 85.71 ± 0.48 |
| TIES | 54.95 ± 1.18 | 50.16 ± 5.58 | **44.35 ± 0.35** | **52.87 ± 1.61** | 32.48 ± 1.11 | 46.66 ± 3.37 | 71.34 ± 1.57 | **86.81 ± 0.23** |
| Task Arithmetic | 18.06 ± 12.38 | 2.73 ± 2.36 | 13.61 ± 11.02 | 16.68 ± 15.07 | 26.22 ± 1.74 | 3.30 ± 2.65 | 28.67 ± 24.04 | 35.23 ± 31.08 |
| Model Stock | 37.79 ± 2.26 | 2.27 ± 0.96 | 34.36 ± 2.60 | 38.29 ± 1.26 | 32.59 ± 0.23 | **54.24 ± 0.20** | 59.19 ± 2.08 | 43.57 ± 11.01 |
| Ours | **56.01** | **50.39** | 43.49 | 50.75 | **35.22** | 54.13 | **71.77** | 86.33 |
| LLaVA-v1.5-7B | 59.07 | 53.44 | 46.07 | 61.97 | 35.30 | 54.39 | 76.64 | 85.67 |

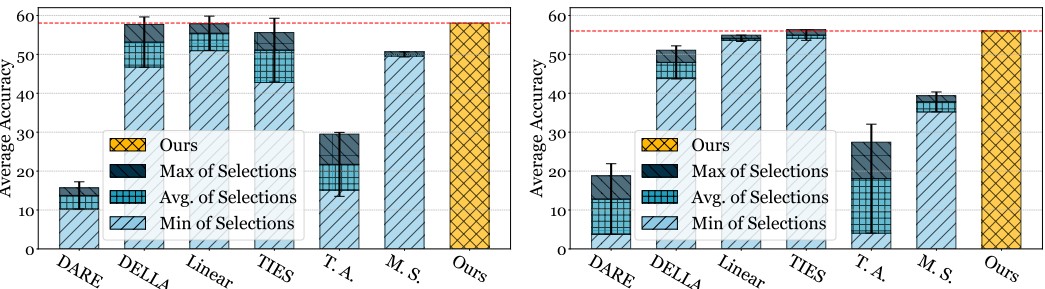

Figure 3: Comparison of `UQ-Merge` against minimum, average, and maximum performance of baselines in random selections on held-in (left) and held-out (right) datasets. T. A. and M. S. stand for Task Arithmetic and Model Stock. Error bar represents the 95% confidence interval.

### 4.3 UQ GUIDED MODEL MERGING SURPASSES EXISTING MERGING METHODS

We further compare `UQ-Merge` in a more challenging setting, where baseline merging methods have an "unfair" advantage to access all the knowledge within models and merge. Experiment results show that ❶ Compared to baseline methods, `UQ-Merge` still achieves the best average accuracy on both held-in and held-out datasets, surpassing these single-modal methods by 0.54% ∼ 51.69% and 1.3% ∼ 52.6% respectively. ❷ Compared to the average performance of merging randomly selected portions of all models, all merging methods except DARE and Task Arithmetic enjoy performance increase by adding more models. This suggests the benefit of incorporating more models from diverse tasks to build a stronger model and supports our claim that model merging is a cheap way to aggregate knowledge from different models. It is worth noting that existing single-modal merging methods have a certain ability to resolve potentially harmful models when merging, by adopting model weight level manipulation to resolve weight conflict and preserve knowledge from different tasks (Ilharco et al., 2023; Yadav et al., 2023; Yu et al., 2024). However, these methods are limited to single-modal model merging. Our `UQ-Merge` is orthogonal to these works, as we consider model level removal of harmful over-confident models in MLLM merging situations to improve the performance, and can benefit from the techniques to ameliorate weight conflict.

### 4.4 RESEARCH QUESTIONS AND ABLATION STUDY

In this section, we conduct an in-depth investigation of the designs adopted in `UQ-Merge` and how they contribute to improved performance. Specifically, we address the following: (1) Is UQ a more

Table 2: The comparison between `UQ-Merge`, single-modal merging methods and LLaVA-v1.5. **Baselines merge all 10 models.** The best and second-best are in **bold** and underline.

| Merging Methods | Vision-Language Classification Datasets | | | | | |
| --- | --- | --- | --- | --- | --- | --- |
| | Average | AI2D | ScienceQA | SeedBench | MMBench | OOD-CV |
| DARE | 6.36 | 0.00 | 14.10 | 17.54 | 14.10 | 17.54 |
| DELLA | 57.50 | 62.96 | **71.69** | **41.94** | **71.69** | **41.94** |
| Linear | 56.90 | 67.39 | 69.98 | 36.93 | 69.98 | 36.93 |
| TIES | 57.51 | 62.99 | **71.69** | **41.94** | **71.69** | **41.94** |
| Task Arithmetic | 10.94 | 0.29 | 26.58 | 25.78 | 26.58 | 25.78 |
| Model Stock | 51.55 | 65.21 | 65.74 | 29.47 | 65.74 | 29.47 |
| Ours | **58.05** | **68.07** | 70.35 | 39.52 | 70.35 | 39.52 |
| LLaVA-v1.5-7B | 59.07 | 46.07 | 35.30 | 54.39 | 76.64 | 85.67 |

| Merging Methods | Vision-Language Generation Datasets | | | | | | | |
| --- | --- | --- | --- | --- | --- | --- | --- | --- |
| | Average | OKVQA | TextVQA | GQA | MMMU | VizWiz | VQAv2 | POPE |
| DARE | 3.41 | 0.02 | 0.23 | 0.00 | 23.56 | 0.00 | 0.03 | 0.00 |
| DELLA | 54.69 | 47.84 | 44.79 | **51.65** | 33.56 | 48.38 | 70.51 | 86.12 |
| Linear | 54.71 | 44.51 | 44.23 | 48.56 | 35.00 | 55.21 | 70.21 | 85.23 |
| TIES | 54.70 | 47.89 | **44.81** | **51.65** | 33.56 | 48.39 | 70.51 | 86.11 |
| Task Arithmetic | 3.56 | 0.00 | 0.04 | 0.00 | 24.89 | 0.01 | 0.01 | 0.00 |
| Model Stock | 45.05 | 8.09 | 38.24 | 42.49 | 32.78 | **56.19** | 63.63 | 73.93 |
| Ours | **56.01** | **50.39** | 43.49 | 50.75 | **35.22** | 54.13 | **71.77** | **86.33** |
| LLaVA-v1.5-7B | 59.07 | 53.44 | 46.07 | 61.97 | 35.30 | 54.39 | 76.64 | 85.67 |

Table 3: Comparison of uncertainty and accuracy as different guidance. Each guidance is implemented with ascending and descending orders to sort models. Accuracy for merging guidance is tested on held-in datasets, and all results are reported on held-out datasets. The best and second-best performances are highlighted in **bold** and underline.

| Guidance | Order | Performance with Different Guidance | | | | | | | |
| --- | --- | --- | --- | --- | --- | --- | --- | --- | --- |
| | | Average | OKVQA | TextVQA | GQA | MMMU | VizWiz | VQAv2 | POPE |
| Uncertainty | Ascending | 54.30 | 45.64 | **44.61** | 45.31 | 34.56 | 54.52 | 70.05 | 85.42 |
| | Descending | **56.01** | **50.39** | 43.49 | **50.75** | 35.22 | 54.13 | **71.77** | **86.33** |
| Accuracy | Ascending | 54.72 | 44.56 | 44.12 | 48.58 | 35.00 | 55.34 | 70.22 | 85.24 |
| | Descending | 52.27 | 30.33 | 40.68 | 50.58 | **35.67** | 55.73 | 67.34 | 85.58 |

effective way to exclude harmful models, and how should uncertainty of models be used? (Section 4.4.1) (2) How to select models to merge after quantifying models' uncertainty? (Section 4.4.2) (3) How to design UQ? (Section 4.4.3) (4) After selection, how to merge models? (Section 4.4.4).

### 4.4.1 RQ1: IS UQ MORE EFFECTIVE THAN ACCURACY? HOW TO USE UNCERTAINTY? A1: YES; SORT BY DECREASING UNCERTAINTY

In `UQ-Merge`, uncertainty is adopted to measure each model and exclude harmful models. An intuitive alternative to uncertainty is the accuracy of the model on validation datasets, with sorting done in either ascending or descending order. To address these research questions, we compare uncertainty and accuracy to determine which serves as better guidance. We replace uncertainty in `UQ-Merge` with accuracy on held-in datasets and test both kinds of guidance in ascending and descending order. Other components in `UQ-Merge` are kept untouched. We evaluate these modified designs on held-out datasets due to the usage of held-in datasets for testing accuracy. As shown in Tab 3, sorting by descending uncertainty achieves the best average performance compared to other options, confirming the effectiveness of our design. Compared to ascending uncertainty, descending order leads to a better performance, which justifies our aim to exclude over-confident models.

### 4.4.2 RQ2: HOW TO SELECT MODELS TO MERGE? A2: WHEN THE UNCERTAINTY OF THE MERGED MODEL IS THE LOWEST

After sorting models by descending order of uncertainty, it remains unsure how to exclude harmful models and select beneficial ones. In `UQ-Merge`, this process is conducted by picking the merged

Table 4: The correlation among uncertainty, average accuracy on validation benchmarks, and average accuracy on held-out benchmarks. The lowest uncertainty, the highest validation accuracy and the highest generation accuracy are marked in **bold**.

| # Models | 1 | 2 | 3 | 4 | 5 | 6 | 7 | 8 | 9 | 10 |
|---|---|---|---|---|---|---|---|---|---|---|
| Uncertainty | 0.21197 | 0.13290 | 0.05609 | 0.04866 | 0.04531 | 0.04155 | **0.03950** | 0.03954 | 0.03740 | 0.03675 |
| Validation Accuracy | 18.75 | 18.48 | 36.32 | 37.30 | 38.84 | 39.20 | **39.91** | 39.15 | 39.28 | 39.60 |
| Held-out Accuracy | 47.09 | 51.22 | 52.47 | 54.15 | 54.13 | 54.54 | **56.01** | 55.56 | 54.15 | 54.14 |

model that has the lowest uncertainty. To verify this, we evaluated the correlation between uncertainty, accuracy on validation benchmarks, and accuracy on held-out datasets. In our experiments, we used RealWorldQA (xAI, 2024), Seedbench 2 Plus (Li et al., 2024a), and OcrBench (Liu et al., 2024c), which focus on real-world QA, multi-disciplinary knowledge, and text recognition, respectively. As shown in Table 4, the lowest uncertainty and highest validation accuracy align with the peak performance on held-out datasets. Our findings indicate that lower uncertainty corresponds to better performance and supports our design. We attribute this to the enhanced capability of the merged model that makes it more robust to input perturbation and could generate consistent answers.

### 4.4.3 RQ3: HOW TO DESIGN PERTURBATION? A3: VISION-LANGUAGE INPUT PERTURBATION IS CRUCIAL

In this research question, we aim to investigate how different perturbation designs would affect the merging performance of `UQ-Merge`. Specifically, we compare the input and model perturbation method adopted in `UQ-Merge` versus only using input perturbation, by using them as different UQ functions in our `UQ-Merge` framework and test the merged model. We implement input perturbation following the same design of `UQ-Merge`, by adding random image transformations and text prompts to the image and text branches respectively. As shown in Table 5, when only use input perturbation, the performance is slightly improved on held-in datasets. On held-out datasets, the performance is slightly worse for input perturbation only. We attribute this to the robust capability of LLM backbones and dynamic sparsity of LLM inference (Liu et al., 2023b), which makes model perturbation may not significantly affect the performance of the LLM backbone.

Table 5: Comparison of perturbation types. Results are average accuracy on datasets.

| | Held-in | Held-out |
|---|---|---|
| Input | **56.82** | 56.00 |
| Input & Model | 56.80 | **56.01** |

### 4.4.4 RQ4: WHAT MERGING METHOD TO USE GIVEN A GROUP OF MLLMS? A4: TIES, LINEAR OR DELLA

Existing merging methods are designed to deal with single-modal merging, and it remains unclear how these merging methods perform for merging multimodal models. In this research question, we explore the performance of these single-modal merging methods in the multimodal scenario by evaluating their performance on a given group of models. Specifically, we evaluate DARE, DELLA, Linear, TIES, Task Arithmetic, and Model Stock on held-in and held-out datasets and calculate the average performance on all the datasets. From results in Table 6 we observe that DELLA, Linear, and TIES perform better than other methods. In 10-model merging, all instruction-tuned models are merged. As shown in Table 6, given the same ten models, TIES achieve the best performance.

Table 6: Comparison of merging methods on the same group of models. Results are average accuracy on held-in and held-out datasets.

| Merging Methods | Number of Models | |
|---|---|---|
| | 7 | 10 |
| DARE | 22.30 | 4.64 |
| DELLA | 56.26 | 55.86 |
| Linear | **56.86** | 55.62 |
| TIES | 56.26 | **55.87** |
| Task Arithmetic | 22.21 | 6.64 |
| Model Stock | 46.82 | 47.76 |

When merging seven models with the highest uncertainty, we observe that ❶ The performance of all merging methods improved, demonstrating the benefit of model selection. ❷ linear merging achieves the best performance, which supports our choice in `UQ-Merge` that linearly merges models.

## 5 CONCLUSION

In this paper, we present a novel MLLM merging algorithm `UQ-Merge` to aggregate diverse knowledge of models into a single MLLM. We design a vision-language perturbation-based UQ and em-

ploy it to guide the merging process. As a result, `UQ-Merge` could identify beneficial models to merge and use the uncertainty value to decide the merging order and number of models to merge, and apply appropriate merging on selected models. Extensive experiments on datasets from diverse domains consistently demonstrate the effective model selection and significantly improved performance of our algorithm. Future works include the extension to more multimodal models and tasks like audio-language models.

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

# A ADDITIONAL EXPERIMENT RESULTS

## A.1 EVALUATION OF FINE-TUNED MODELS

Table 7: Uncertainty of all models on held-in datasets.

| Tuning Dataset | Average | AI2D | MMBench | OOD-CV | ScienceQA | SeedBench |
|---|---|---|---|---|---|---|
| OKVQA | 0.1314 | 0.1199 | 0.1073 | 0.1350 | 0.1149 | 0.1802 |
| OCRVQA | 0.0548 | 0.0555 | 0.0564 | 0.0669 | 0.0505 | 0.0446 |
| GQA | 0.2141 | 0.2032 | 0.2375 | 0.2136 | 0.2219 | 0.1941 |
| VQAv2 | 0.0322 | 0.0286 | 0.0355 | 0.0350 | 0.0311 | 0.0306 |
| TextCaps | 0.0073 | 0.0078 | 0.0085 | 0.0115 | 0.0054 | 0.0030 |
| A_OKVQA | 0.1078 | 0.1318 | 0.0746 | 0.1066 | 0.1132 | 0.1130 |
| RefCOCO | 0.0275 | 0.0250 | 0.0309 | 0.0311 | 0.0267 | 0.0238 |
| LLaVA-Instruct | 0.0713 | 0.0670 | 0.0767 | 0.0787 | 0.0672 | 0.0668 |
| ShareGPT | 0.0317 | 0.0246 | 0.0380 | 0.0386 | 0.0320 | 0.0255 |
| VG | 0.0379 | 0.0334 | 0.0412 | 0.0417 | 0.0360 | 0.0372 |
| Tuning Dataset | Average | AI2D | MMBench | OOD-CV | ScienceQA | SeedBench |
| OKVQA | 0.1305 | 0.1199 | 0.1072 | 0.1317 | 0.1152 | 0.1787 |
| OCRVQA | 0.0544 | 0.0550 | 0.0560 | 0.0663 | 0.0505 | 0.0444 |
| GQA | 0.2141 | 0.2024 | 0.2408 | 0.2084 | 0.2229 | 0.1962 |
| VQAv2 | 0.0323 | 0.0288 | 0.0361 | 0.0347 | 0.0312 | 0.0307 |
| TextCaps | 0.0072 | 0.0077 | 0.0087 | 0.0114 | 0.0055 | 0.0029 |
| A_OKVQA | 0.1077 | 0.1326 | 0.0751 | 0.1070 | 0.1117 | 0.1123 |
| RefCOCO | 0.0273 | 0.0251 | 0.0305 | 0.0305 | 0.0265 | 0.0238 |
| LLaVA-Instruct | 0.0713 | 0.0668 | 0.0771 | 0.0803 | 0.0662 | 0.0659 |
| ShareGPT | 0.0318 | 0.0246 | 0.0375 | 0.0394 | 0.0318 | 0.0255 |
| VG | 0.0378 | 0.0335 | 0.0408 | 0.0419 | 0.0361 | 0.0369 |
| Tuning Dataset | Average | AI2D | MMBench | OOD-CV | ScienceQA | SeedBench |
| OKVQA | 0.1306 | 0.1205 | 0.1067 | 0.1334 | 0.1157 | 0.1767 |
| OCRVQA | 0.0547 | 0.0554 | 0.0562 | 0.0668 | 0.0501 | 0.0448 |
| GQA | 0.2130 | 0.2020 | 0.2369 | 0.2115 | 0.2220 | 0.1926 |
| VQAv2 | 0.0323 | 0.0288 | 0.0357 | 0.0344 | 0.0313 | 0.0311 |
| TextCaps | 0.0073 | 0.0078 | 0.0085 | 0.0115 | 0.0055 | 0.0030 |
| A_OKVQA | 0.1082 | 0.1324 | 0.0758 | 0.1069 | 0.1123 | 0.1135 |
| RefCOCO | 0.0273 | 0.0249 | 0.0303 | 0.0308 | 0.0266 | 0.0238 |
| LLaVA-Instruct | 0.0713 | 0.0666 | 0.0763 | 0.0801 | 0.0667 | 0.0666 |
| ShareGPT | 0.0316 | 0.0245 | 0.0379 | 0.0388 | 0.0319 | 0.0247 |
| VG | 0.0376 | 0.0335 | 0.0406 | 0.0417 | 0.0360 | 0.0363 |

In Table 7, we provide uncertainty quantification results of fine-tuned models on held-in datasets. We conduct evaluation three times and the final uncertainty is the average. As observed in Table 7, the uncertainty is stable and consistent, showcasing the effectiveness and stability of our vision-language perturbation-based UQ.

## A.2 EVALUATION OF BASELINES OVER RANDOM SELECTIONS

Table 8: Accuracy of baselines on held-in datasets when merging random model selections.

| Merging Methods | Vision-Language Classification Datasets | | | | | |
|---|---|---|---|---|---|---|
| | Average | AI2D | ScienceQA | SeedBench | MMBench | OOD-CV |
| DARE | 15.68 | 11.98 | 5.19 | 8.47 | 26.53 | 26.21 |
| DELLA | 46.73 | 34.42 | 50.05 | 46.97 | 68.76 | 33.45 |
| Linear | 57.85 | 51.65 | 67.83 | 60.25 | 70.19 | 39.35 |
| TIES | 55.60 | 49.29 | 55.96 | 56.14 | 70.74 | 45.85 |
| Task Arithmetic | 15.13 | 5.18 | 12.98 | 5.20 | 26.53 | 25.78 |
| Model Stock | 50.65 | 44.82 | 65.12 | 50.08 | 64.86 | 28.37 |
| Merging Methods | Average | AI2D | ScienceQA | SeedBench | MMBench | OOD-CV |
| DARE | 10.18 | 0.00 | 0.00 | 0.00 | 25.40 | 25.50 |
| DELLA | 55.02 | 47.54 | 57.70 | 56.57 | 70.81 | 42.47 |
| Linear | 57.48 | 51.36 | 67.90 | 59.91 | 70.17 | 38.07 |
| TIES | 54.95 | 46.76 | 56.78 | 57.49 | 70.90 | 42.83 |
| Task Arithmetic | 20.54 | 24.19 | 0.00 | 24.96 | 27.46 | 26.10 |
| Model Stock | 49.56 | 43.56 | 65.10 | 48.56 | 63.24 | 27.34 |
| Merging Methods | Average | AI2D | ScienceQA | SeedBench | MMBench | OOD-CV |
| DARE | 15.39 | 12.05 | 2.96 | 8.72 | 26.81 | 26.42 |
| DELLA | 57.69 | 47.73 | 66.16 | 61.86 | 71.20 | 41.51 |
| Linear | 50.95 | 45.21 | 56.73 | 51.31 | 69.10 | 32.39 |
| TIES | 42.77 | 34.68 | 44.47 | 32.11 | 68.41 | 34.16 |
| Task Arithmetic | 29.52 | 25.29 | 38.54 | 30.89 | 28.80 | 24.08 |
| Model Stock | 49.77 | 43.04 | 65.00 | 48.70 | 64.21 | 27.88 |
| Ours | 58.05 | 51.75 | 68.07 | 60.56 | 70.35 | 39.52 |

Table 9: Accuracy of baselines on held-out datasets when merging random model selections.

| Merging Methods | Vision-Language Generation Datasets | | | | | | | |
| --- | --- | --- | --- | --- | --- | --- | --- | --- |
| | Average | OKVQA | TextVQA | GQA | MMMU | VizWiz | VQAv2 | POPE |
| DARE | 18.78 | 0.41 | 7.82 | 19.06 | 27.11 | 3.85 | 32.48 | 40.70 |
| DELLA | 43.78 | 41.69 | 39.74 | 40.83 | 31.67 | 40.75 | 62.46 | 49.32 |
| Linear | 53.78 | 39.45 | 40.67 | 49.65 | 34.67 | 57.20 | 69.01 | 85.83 |
| TIES | 56.31 | 56.08 | 44.75 | 54.72 | 33.33 | 45.32 | 73.03 | 86.94 |
| Task Arithmetic | 4.04 | 0.04 | 0.91 | 0.33 | 24.67 | 0.26 | 0.96 | 1.09 |
| Model Stock | 39.37 | 3.35 | 37.30 | 39.54 | 32.33 | 54.38 | 61.51 | 47.18 |
| Merging Methods | Average | OKVQA | TextVQA | GQA | MMMU | VizWiz | VQAv2 | POPE |
| DARE | 3.71 | 0.00 | 0.00 | 0.00 | 26.00 | 0.00 | 0.00 | 0.00 |
| DELLA | 51.08 | 47.94 | 43.15 | 43.70 | 32.44 | 36.94 | 69.17 | 84.23 |
| Linear | 53.66 | 43.35 | 43.43 | 49.02 | 33.67 | 52.88 | 68.06 | 85.18 |
| TIES | 54.15 | 49.39 | 44.19 | 52.08 | 31.22 | 44.17 | 71.07 | 86.94 |
| Task Arithmetic | 22.70 | 4.47 | 20.60 | 19.69 | 25.89 | 4.53 | 41.02 | 42.72 |
| Model Stock | 38.79 | 1.95 | 33.38 | 38.30 | 32.67 | 54.34 | 58.55 | 52.33 |
| Merging Methods | Average | OKVQA | TextVQA | GQA | MMMU | VizWiz | VQAv2 | POPE |
| DARE | 15.99 | 0.70 | 8.61 | 13.66 | 25.56 | 3.44 | 26.84 | 33.10 |
| DELLA | 48.95 | 31.97 | 40.62 | 47.34 | 34.22 | 48.07 | 64.35 | 76.06 |
| Linear | 54.96 | 51.04 | 44.12 | 50.83 | 32.78 | 48.33 | 71.48 | 86.11 |
| TIES | 54.40 | 45.01 | 44.11 | 51.81 | 32.89 | 50.50 | 69.93 | 86.54 |
| Task Arithmetic | 27.45 | 3.67 | 19.32 | 30.01 | 28.11 | 5.11 | 44.02 | 61.88 |
| Model Stock | 35.20 | 1.51 | 32.39 | 37.02 | 32.78 | 54.01 | 57.50 | 31.21 |
| Ours | 56.01 | 50.39 | 43.49 | 50.75 | 35.22 | 54.13 | 71.77 | 86.33 |

In Table 8 and Table 9, we provide performance of baseline single-modal merging methods on held-in and held-out datasets over model selections. As observed in tables, DELLA, Linear and TIES consistently outperform other merging methods with a small variance. The average and standard error are reported based on results above, and the error bar represents the 95% confidence interval.

## A.3 EVALUATION DURING MERGING STEPS

Table 10: Accuracy on validation datasets during merging steps.

| Validation Datasets | Merging Steps | | | | | | | | | |
| --- | --- | --- | --- | --- | --- | --- | --- | --- | --- | --- |
| | 1 | 2 | 3 | 4 | 5 | 6 | 7 | 8 | 9 | 10 |
| RealWorldQA | 27.58 | 22.48 | 47.45 | 46.80 | 47.19 | 47.58 | 49.15 | 47.32 | 47.32 | 47.84 |
| SeedBench 2 Plus | 23.36 | 10.36 | 39.00 | 40.40 | 41.33 | 41.81 | 42.07 | 41.72 | 42.51 | 42.16 |
| OCRBench | 5.30 | 22.60 | 22.50 | 24.70 | 28.00 | 28.20 | 28.50 | 28.40 | 28.00 | 28.80 |

Table 11: Uncertainty on held-in datasets during merging steps.

| Held-in Datasets | Merging Steps | | | | | | | | | |
| --- | --- | --- | --- | --- | --- | --- | --- | --- | --- | --- |
| | 1 | 2 | 3 | 4 | 5 | 6 | 7 | 8 | 9 | 10 |
| AI2D | 0.201288 | 0.116523 | 0.068064 | 0.061422 | 0.057592 | 0.050615 | 0.046001 | 0.046755 | 0.043276 | 0.042478 |
| MMBench | 0.234769 | 0.118624 | 0.045755 | 0.039847 | 0.037836 | 0.035598 | 0.035828 | 0.034876 | 0.033679 | 0.033072 |
| ScienceQA | 0.211437 | 0.136477 | 0.066517 | 0.058877 | 0.053063 | 0.049049 | 0.045427 | 0.046717 | 0.043556 | 0.042217 |
| SeedBench | 0.221570 | 0.126527 | 0.055429 | 0.047013 | 0.043838 | 0.040794 | 0.038647 | 0.038368 | 0.037026 | 0.036064 |
| OOD-CV | 0.190767 | 0.166342 | 0.044704 | 0.036159 | 0.034233 | 0.031680 | 0.031603 | 0.030970 | 0.029448 | 0.029895 |

In Table 11, Table 10 and Table 12, we present the evaluation results during merging steps.

Table 12: Accuracy on held-out datasets during merging steps.

| Held-out Datasets | Merging Steps | | | | | | | | | |
|---|---|---|---|---|---|---|---|---|---|---|
| | 1 | 2 | 3 | 4 | 5 | 6 | 7 | 8 | 9 | 10 |
| OKVQA | 27.05 | 54.77 | 50.48 | 48.01 | 46.21 | 44.13 | 50.39 | 47.90 | 48.01 | 46.21 |
| TextVQA | 23.15 | 36.06 | 34.24 | 37.96 | 41.19 | 41.82 | 43.49 | 43.22 | 37.96 | 41.19 |
| GQA | 61.73 | 54.08 | 50.77 | 50.29 | 48.97 | 49.98 | 50.75 | 49.75 | 50.29 | 48.97 |
| MMMU | 30.00 | 29.78 | 33.78 | 34.22 | 34.44 | 35.00 | 35.22 | 35.22 | 34.22 | 34.56 |
| VizWiz | 44.25 | 34.69 | 46.72 | 54.36 | 54.61 | 55.66 | 54.13 | 55.31 | 54.36 | 54.61 |
| VQAv2 | 60.03 | 66.09 | 66.70 | 68.30 | 67.81 | 68.71 | 71.77 | 71.25 | 68.30 | 67.81 |
| POPE | 83.39 | 83.07 | 84.60 | 85.91 | 85.68 | 86.46 | 86.33 | 86.29 | 85.91 | 85.66 |

Table 13: The comparison between `UQ-Merge`, single-modal merging methods and LLaVA-v1.5 that trains on the combined dataset. **Baseline methods merge randomly selected the same number of models to `UQ-Merge`.** Average and standard error of the accuracy of baselines across selections are reported. Results are measured with 3 selections. The best and second-best performances are highlighted in **bold** and underline, respectively.

| Vision-Language Classification Datasets | | | | | | |
|---|---|---|---|---|---|---|
| Merging Methods | Average | AI2D | ScienceQA | SeedBench | MMBench | OOD-CV |
| | 13.75 ± 3.09 | 8.01 ± 6.94 | 2.72 ± 2.60 | 5.73 ± 4.96 | 26.25 ± 0.75 | 26.04 ± 0.48 |
| DELLA | 53.15 ± 5.72 | 43.23 ± 7.63 | 57.97 ± 8.06 | 55.13 ± 7.55 | 70.26 ± 1.31 | 39.14 ± 4.95 |
| Linear | 55.43 ± 3.88 | 49.41 ± 3.64 | 64.15 ± 6.43 | 57.16 ± 5.07 | 69.82 ± 0.62 | 36.60 ± 3.70 |
| TIES | 51.10 ± 7.23 | 43.58 ± 7.81 | 52.40 ± 6.88 | 48.58 ± 14.28 | 70.02 ± 1.39 | **40.95 ± 6.07** |
| Task Arithmetic | 21.73 ± 7.27 | 18.22 ± 11.31 | 17.17 ± 19.61 | 20.35 ± 13.45 | 27.60 ± 1.14 | 25.32 ± 1.09 |
| Model Stock | 49.99 ± 0.58 | 43.81 ± 0.92 | 65.07 ± 0.06 | 49.11 ± 0.84 | 64.10 ± 0.82 | 27.86 ± 0.52 |
| Ours | **58.05** | **51.75** | **68.07** | **60.56** | **70.35** | 39.52 |
| LLaVA-v1.5-7B | 64.96 | 54.79 | 70.43 | 60.49 | 72.04 | 67.05 |

| Merging Methods | Vision-Language Generation Datasets | | | | | | |
|---|---|---|---|---|---|---|---|
| | Average | OKVQA | TextVQA | GQA | MMMU | VizWiz | VQAv2 | POPE |
| DARE | 12.83 ± 8.01 | 0.37 ± 0.35 | 5.48 ± 4.76 | 10.91 ± 9.82 | 26.22 ± 0.80 | 2.43 ± 2.11 | 19.77 ± 17.35 | 24.60 ± 21.64 |
| DELLA | 47.94 ± 3.75 | 40.53 ± 8.05 | 41.17 ± 1.77 | 43.96 ± 3.26 | 32.78 ± 1.31 | 41.92 ± 5.66 | 65.33 ± 3.46 | 69.87 ± 18.26 |
| Linear | 54.13 ± 0.72 | 44.61 ± 5.90 | 42.74 ± 1.83 | 49.83 ± 0.92 | 33.71 ± 0.95 | 52.80 ± 4.44 | 69.52 ± 1.77 | 85.71 ± 0.48 |
| TIES | 54.95 ± 1.18 | 50.16 ± 5.58 | **44.35 ± 0.35** | **52.87 ± 1.61** | 32.48 ± 1.11 | 46.66 ± 3.37 | 71.34 ± 1.57 | **86.81 ± 0.23** |
| Task Arithmetic | 18.06 ± 12.38 | 2.73 ± 2.36 | 13.61 ± 11.02 | 16.68 ± 15.07 | 26.22 ± 1.74 | 3.30 ± 2.65 | 28.67 ± 24.04 | 35.23 ± 31.08 |
| Model Stock | 37.79 ± 2.26 | 2.27 ± 0.96 | 34.36 ± 2.60 | 38.29 ± 1.26 | 32.59 ± 0.23 | **54.24 ± 0.20** | 59.19 ± 2.08 | 43.57 ± 11.01 |
| Ours | **56.01** | **50.39** | 43.49 | 50.75 | **35.22** | 54.13 | **71.77** | 86.33 |
| LLaVA-v1.5-7B | 59.07 | 53.44 | 46.07 | 61.97 | 35.30 | 54.39 | 76.64 | 85.67 |

# B MORE IMPLEMENTATIIN DETAILS

## B.1 TEXT PERTURBATION

Prompts used for perturbation of text inputs:

- 'you are a helpful assistant',

- 'you are a question-answering assistant',

- 'you are a nice assistant',

- 'You are a helpful assistant',

- 'You are a question-answering assistant',

- 'You are a nice assistant',

- 'You are a helpful assistant.',

- 'You are a question-answering assistant.',

- 'You are a nice assistant.'

### B.2 IMAGE PERTURBATION

The image perturbation is implemented by utilizing the implementation of AugMix (Hendrycks et al., 2020) in torchvision (Aug), and all parameters are set to default.

### B.3 DATASETS FOR UNCERTAINTY QUANTIFICATION

We use the code from (Kostumov et al., 2024) to process MMBench (Liu et al., 2023a), OODCV-VQA (Zhao et al., 2022), ScienceQA (Lu et al., 2022), SEEDBench (Li et al., 2023a), and AI2D (Kembhavi et al., 2016) for vision-language perturbation-based UQ.

### B.4 EVALUATION OF MODELS

We adopt LMMs-Eval (Zhang et al., 2024b) to conduct evaluation of models on all benchmarks except MMBench and OODCV-VQA, which are evaluated directly using our pre-processed datasets.

