# OpenReview forum: "$\texttt{UQ-Merge}$: UNCERTAINTY GUIDED MULTIMODAL LARGE LANGUAGE MODEL MERGING"
_ICLR.cc/2025/Conference — ICLR 2025 Conference Withdrawn Submission_

### Official Review · Reviewer_8odd · 2024-10-24

**Soundness:** 3
**Presentation:** 2
**Contribution:** 3
**Rating:** 6
**Confidence:** 3

**Summary:**

This paper proposes a novel approach (UQ-Merge) for merging MLLMs (i.e. LVLMs) using uncertainty quantification (UQ) to enhance model performance. The key challenge addressed is the suboptimal performance of existing model merging methods due to the inclusion of over-confident models and the lack of vision-language-specific merging designs. Therefore, they propose a strategy that uses perturbation to detect over-confidence, ranks each candidate model according to confidence, and ensures the final output merged model has the lowest uncertainty.

**Strengths:**

1. Their proposed UQ-merge algorithm, which selects beneficial models by evaluating uncertainty and excludes over-confident models, and merges models in a selective order based on their uncertainty scores, thereby improving the overall performance of the final merged model, is an intuitive and useful strategy.
2. Since model merge strategies for MLLMs are underexplored, this paper has promising.
3. The experiments clearly demonstrate the effectiveness of this strategy.
4. While existing merge strategies, including those for LLMs, typically rely on superficial performance metrics, this paper shows in section 4.4.1 that selecting models based on the implicit metric of uncertainty yields better results.

**Weaknesses:**

1. In my opinion, the main weakness lies in the question of scalability for real-world applications. First, real-world MLLM usage is often not task-specific, and evaluation test sets are not provided in large quantities or good quality. From this perspective, it would be valuable to discuss how an algorithm that measures uncertainty on specific refined datasets could be applied to real-world unseen datasets in the future or to discuss evaluation using limited test sets.
2. Measuring the uncertainty of all candidate models and repeatedly measuring the uncertainty of merged models to find good merge combinations for a single task-specific training set is very computationally costly. This will become an increasingly critical drawback as dataset quality and scale increase, and MLLM sizes grow.
3. The core of this algorithm is 'uncertainty measurement.' It heavily relies on the perturbation method, and while section 4.4.3 states 'VISION-LANGUAGE INPUT PERTURBATION IS CRUCIAL', it's disappointing that they didn't create and analyze a variety of strategies for this.

**Questions:**

Were there various attempts at methods for measuring uncertainty during this project? It seems like it could have been possible to try approaches other than perturbation or to experiment with different perturbation methods. A discussion about this could be a very interesting and important contribution.

---

> ### Author Response · Authors · 2024-11-19
> **Point-to-point Response to Reviewer 8odd**
>
> We sincerely appreciate your careful review and a great summary of our contributions as **intuitive, useful, promising and effective**. And thank you very much for your construction comments. We provide our response as follows:
>
> **Q1: Discussion of evaluation using limited test sets**
>
> **A1:** Thanks for your valuable suggestions. We present our results with limited datasets for UQ:
>
> |            | Average | AI2D  | ScienceQA | OKVQA | POPE  |
> | ---------- | :-----: | :---: | :-------: | :---: | :---: |
> | Limited UQ |  64.16  | 51.70 |   67.95   | 50.71 | 86.26 |
> | Ours       |  64.14  | 51.75 |   68.07   | 50.39 | 86.33 |
>
> From the results above we can see that the performance between full or limited datasets for UQ is very close. These results validate our design and prove its data efficiency.
>
> We further tested our method with limited evaluation data from [1], and the results are shown below, where our method still performs better compared to baselines.
>
> |        |  Average  |   AI2D    |   OKVQA   |    GQA    |   VQAv2   |
> | ------ | :-------: | :-------: | :-------: | :-------: | :-------: |
> | Linear |   54.54   |   52.00   |   45.12   |   56.00   |   65.02   |
> | TIES   |   55.44   |   50.20   |   48.00   |   58.20   |   65.34   |
> | Ours   | **57.69** | **54.40** | **51.20** | **58.60** | **66.56** |
>
>
> > [1] Kaichen Zhang, undefined., et al, "LMMs-Eval: Reality Check on the Evaluation of Large Multimodal Models," 2024.
>
> **Q2: Computationally cost of UQ**
>
> **A2:** Thank you for the question. Indeed the running time of our method will increase as the dataset and MLLM enlarge. But our method is a **one-time** effort and the overhead (5h for a single model on a RTX A6000 Ada) will be amortized by the deployment and the performance benefits make the effort worthwhile.
>
> **Q3: Ablation of different perturbation strategies**
>
> **A3:** Thank you for the great question. We tested our method with three perturbation variants (image-only, text-only, and conformal prediction [2] as the UQ). The results are below:
>
> |                      |  Average  |   AI2D    | ScienceQA |   OKVQA   |   POPE    |
> | -------------------- | :-------: | :-------: | :-------: | :-------: | :-------: |
> | Text Only            |   61.93   |   50.55   |   67.39   |   44.55   |   85.23   |
> | Image Only           |   61.88   |   50.58   |   67.51   |   44.35   |   85.07   |
> | Conformal Prediction |   61.81   |   50.42   |   67.19   |   44.27   |   85.37   |
> | Ours                 | **64.14** | **51.75** | **68.07** | **50.39** | **86.33** |
>
> The results demonstrates the effectiveness of our design that utilizes both image and text perturbation to evaluate models.
>
> > [2] Vasily Kostumov, undefined., et al, "Uncertainty-Aware Evaluation for Vision-Language Models," 2024.

---

> > ### Comment · Reviewer_8odd · 2024-11-20
> >
> > The discussion still remains insufficient regarding how to effectively address the limited test set issue. The extent of the test set's limitations is not specified, and it seems difficult to add performance difference experiments based on various ratios to the main text as a minor revision. The description of perturbation strategies is also inadequate, and I don't believe the main weaknesses have been additionally resolved. However, I will maintain my score as the paper's contributions are still worthy of acknowledgment.

---

> > > ### Author Response · Authors · 2024-11-20
> > >
> > > Thank you for your rapid response and sorry for all the confusion.
> > >
> > > **Settings of limited datasets**
> > >
> > > For the limited datasets of UQ, we used 10% samples of the full dataset, and the numbers of samples used in this setting are:
> > >
> > > |           | MMBench | OOD-CV | ScienceQA | SeedBench | AI2D |
> > > | --------- | :-----: | :----: | :-------: | :-------: | :--: |
> > > | # Samples |   438   |  284   |    395    |   1423    | 1550 |
> > >
> > > We are now running experiments that use 5%, 20% and 30% samples of the full datasets and will report the results within two days. For experiments with limited evaluation datasets, we used 500 samples for each of AI2D, OKVQA, GQA and VQAv2.
> > >
> > > **Description of perturbation**
> > >
> > > We followed our perturbation design in Section 3.3 and created the image/text only settings. In the image-only one we only perturbed input images by applying random composition of image transformation functions such as Shear, Translate, Rotate, Equalize, and Posterize from [1]. For text-only perturbation, we prepended random system prompts like "you are a helpful assistant" to the input question. All the system prompts are listed in Section B.1. For conformal prediction as an alternative UQ method, we utilized [2] to measure the Set Size (the larger the more uncertainty) on MMBench, OOD-CV, ScienceQA, SeedBench and AI2D.
> > >
> > > > [1] Hendrycks, D., et al. "Augmix: A simple data processing method to improve robustness and uncertainty," in *arXiv preprint arXiv:1912.02781*, 2019. \
> > > > [2] Vasily Kostumov, undefined., et al, "Uncertainty-Aware Evaluation for Vision-Language Models," 2024.
> > >
> > > **Challenge of scalability**
> > >
> > > For the challenge of scalability, we provided results with 10% UQ samples to show the data efficiency of our method and will report more ratios in two days. And in our experiments in the main text we considered if uncertainty on specific datasets could improve performance on unseen datasets. In the second subtable of Table 1 and Table 2 (Vision-Language Generation Datasets), we tested our method on unseen datasets (not used for UQ), and results show that our method still performs better compared to baselines.

---

> > > > ### Author Response · Authors · 2024-11-21
> > > >
> > > > Please find the results of more ratios for UQ datasets below. **Bold** denotes the best accuracy.
> > > >
> > > > |              |  Average  |   AI2D    | ScienceQA |   OKVQA   |   POPE    |
> > > > | ------------ | :-------: | :-------: | :-------: | :-------: | :-------: |
> > > > | 5% Samples   |   64.14   |   51.75   |   68.07   |   50.40   |   86.33   |
> > > > | 10% Samples  | **64.16** |   51.70   |   67.95   | **50.71** |   86.26   |
> > > > | 20% Samples  |   62.96   | **52.23** | **68.95** |   44.17   | **86.48** |
> > > > | 30% Samples  |   63.96   |   51.42   |   67.85   |   50.37   |   86.18   |
> > > > | No Selection |   61.67   |   49.81   |   67.34   |   44.38   |   85.14   |
> > > >
> > > > In these experiments, we keep the same perturbation design (input perturbation & model perturbation) as described in Section 3.3, and only change the amount of samples used for UQ. As shown in the table, 1) performance of all ratios of samples is better compared to the baseline at the last row where no model selection is considered. 2) Different ratios lead to close performance, showcasing our method’s robustness and efficiency on data usage.

---

> ### Author Response · Authors · 2024-11-24
> **Genuine Request for Follow-up Discussion**
>
> Dear Reviewer 8odd,
>
> We are very grateful for your constructive comments. We have made a substantial effort to respond to your questions. As there are only two days left for the author-reviewer discussion, we sincerely hope that you can provide us feedback before the discussion phase ends, and are happy to answer any follow-up questions.
>
> Best regards, \
> Authors

---

> ### Author Response · Authors · 2024-11-29
> **Thank You & Looking forward to Further Discussion**
>
> Dear Reviewer 8odd,
>
> We appreciate your insightful review and dedication to the review process. Due to the approaching deadline for rebuttal, we would like to respectfully convey our willingness to address any further concerns you may have. In our previous response, we carefully studied your comments and made detailed responses summarized below:
>
> - Further description about the settings of our rebuttal experiments
> - Results on different ratios of samples in the UQ datasets.
>
> Thank you once again for your support.
>
> Warm regards, \
> Authors

---

> ### Author Response · Authors · 2024-11-30
> **Looking forward to Your Reply!**
>
> Dear Reviewer 8odd,
>
> We are very grateful for your constructive comments. We have made a substantial effort in responding to your questions and listed all the paper revisions and additional experiments in the [general response](https://openreview.net/forum?id=SO0manOwUF&noteId=AVlLH4kHW8). As the deadline for rebuttal is approaching, we sincerely hope you can provide us feedback before the discussion phase ends, and we are happy to answer any follow-up questions.
>
> Best regards, \
> Authors

---

> > ### Comment · Reviewer_8odd · 2024-12-01
> >
> > Although I currently have a positive stance, the weaknesses I pointed out were not tested with sufficient detail and were not reflected in the revision. I will maintain my score.

---

> > > ### Author Response · Authors · 2024-12-01
> > > **Thank You for Your Reply!**
> > >
> > > Dear Reviewer 8odd,
> > >
> > > Thank you very much for your rapid response. We sincerely appreciate your positive rating of our work.
> > >
> > > In our response to address your concerns, we conducted:
> > >
> > > - Variations of [perturbation](https://openreview.net/forum?id=SO0manOwUF&noteId=ZDvgihlt7k) and UQ methods and [description](https://openreview.net/forum?id=SO0manOwUF&noteId=ioJgtO5770) for these experiments.
> > > - Effectiveness of our method on [limited test datasets](https://openreview.net/forum?id=SO0manOwUF&noteId=ZDvgihlt7k) for performance evaluation.
> > > - [Various ratios](https://openreview.net/forum?id=SO0manOwUF&noteId=WCiiV9lVj7) of samples for UQ to test the data efficiency of our method.
> > >
> > > If you could kindly point out the details that were lacking in our discussion, we would be more than happy to address your concerns.
> > >
> > > Best regards, \
> > > Authors

---

### Official Review · Reviewer_aFmm · 2024-11-01

**Soundness:** 3
**Presentation:** 3
**Contribution:** 3
**Rating:** 6
**Confidence:** 4

**Summary:**

The paper introduces an uncertainty-guided merging algorithm for MLLMs in vision-language tasks, addressing the limitations of existing merging methods. This approach selectively merges models based on their prediction confidence in both text and vision inputs without needing labeled data. Experiments show substantial accuracy improvements on various benchmarks.

**Strengths:**

- The introduction of an uncertainty-guided merging algorithm offers a novel perspective on aggregating MLLM models, addressing the specific needs of vision-language tasks.
- By avoiding the need for labeled samples in the uncertainty quantification process, the proposed method increases accessibility and applicability across diverse scenarios.

**Weaknesses:**

Overall Evaluation of the Work:
While the study shows promise, the methodology appears to lack a direct connection to multi-modality beyond the use of multi-modal data perturbation. This raises the question of why this approach was adopted within a multi-modal large language model (MLLM) setting, rather than being applied to a broader range of large language model (LLM) tasks. Further clarification on this choice is recommended.

- Prior work, such as that referenced in [1], has already introduced a fusion technique based on uncertainty, focusing specifically on gradient-related uncertainty as opposed to perturbed uncertainty. A detailed discussion comparing these approaches, with attention to their underlying mechanisms and potential advantages, would strengthen the analysis.
- As indicated in [2], some of the collected knowledge may lack essential commonsense elements.
- Furthermore, Itmay be needless to the visual modality in ScienceQA. This raises a concern: might perturbing images in this manner compromise the validity of the visual information?
- The omission of error metrics in Table 1 is notable. Why are those values not reported in Table 1?

[1] Nico et al. Model merging by uncertainty-based gradient matching. In ICLR, 2024.
[2] Chen et al. M3CoT: A Novel Benchmark for Multi-Domain Multi-step Multi-modal Chain-of-Thought. In ACL 2024.

**Questions:**

See weakness for details.

---

> ### Author Response · Authors · 2024-11-19
> **Point-to-point Response to Reviewer aFmm**
>
> Thank you Reviewer aFmm for your comments. we provide pointwise responses below to address your questions:
>
> **Q1: Generalize our method to LLM**
>
> **A1:** Thank you for raising the question. We would like to point out that: Our major motivation is that MLLM merging and UQ for MLLM are research areas less explored, as mentioned by [Reviewer V122](https://openreview.net/forum?id=SO0manOwUF&noteId=vlMPQYmmBX) and [Reviewer 8odd](https://openreview.net/forum?id=SO0manOwUF&noteId=ygOiSaVNuP). So we aim to conduct pioneering efforts in this direction, addressing (1) merging candidate selection and (2) when to stop adding new models. Our design performs modality-related perturbations which also include text perturbation and can be applied to LLM merging. Our future research goal is to generalize UQ-Merge to LLM and other modalities.
>
> **Q2: Comparison with other uncertainty-based merging methods.**
>
> **A2:** Thank you for pointing out this related work. In [1], they provided a model merging method that utilizes **gradient-based** UQ. However we would like to point out that their work is orthogonal to our method as our method aims to aggregate models in a **training-free** manner **without using any labels**, which is more feasible in many real-world applications as acknowledged by you and [Reviewer V122](https://openreview.net/forum?id=SO0manOwUF&noteId=vlMPQYmmBX).
>
> > [1] Nico et al. Model merging by uncertainty-based gradient matching. In ICLR, 2024.
>
> **Q3: Some collected knowledges lack essential commonsense elements**
>
> **A3:** Thank you for the suggestion. Our method aims to aggregate knowledge **from diverse domains** and these models may not all focus on commonsense. In our experiments, A-OKVQA [2] is a dataset used for fine-tuning and SeedBench is used for UQ [3], and both of them are considered with commonsense in Figure 2 of [4]. The model selection from our method includes the model tuned with A-OKVQA, highlighting its recognition for the importance of commonsense. We further removed A-OKVQA tuned model and results are shown below, which demonstrates the importance of commonsense in our model selection.
>
> |             | Average | AI2D  | ScienceQA | OKVQA | POPE  | SeedBench |
> | ----------- | :-----: | :---: | :-------: | :---: | :---: | :-------: |
> | w/ A_OKVQA  |  64.14  | 51.75 |   68.07   | 50.39 | 86.33 |   60.56   |
> | w/o A_OKVQA |  59.16  | 43.69 |   55.07   | 52.56 | 85.3  |   48.32   |
>
> > [2] Chen et al. M3CoT: A Novel Benchmark for Multi-Domain Multi-step Multi-modal Chain-of-Thought. In ACL 2024. \
> > [3] Dustin Schwenk, undefined., et al, "A-OKVQA: A Benchmark for Visual Question Answering using World Knowledge," 2022. \
> > [4] Bohao Li, undefined., et al, "SEED-Bench: Benchmarking Multimodal LLMs with Generative Comprehension," 2023.
>
> **Q4: The necessity of visual modality in ScienceQA**
>
> **A4:** We respectfully disagree with the point that visual modality in ScienceQA may be needless. ScienceQA is a benchmark focusing on multimodal reasoning, which we view as important for an MLLM. While for certain questions the MLLM may only use the text and its knowledge to answer, we can easily find examples like "Which term matches the picture?" in the dataset. In our method, we apply **slight perturbation** to the image to **preserve** the visual information, and in most cases the visual information is still valid.
>
> **Q5: The omission of error metrics in Table 1**
>
> **A5:** Thank you for the question. We included error metrics in Table 1 and think you **may refer to Table 2**. In our experiments we noticed the performance is very stable across runs and baselines always merge all models and have limited randomness. We tested 3 runs under Table 2's setting and the results are below:
>
> |        |     Average      |       AI2D       |    ScienceQA     |      OKVQA       |       POPE       |
> | ------ | :--------------: | :--------------: | :--------------: | :--------------: | :--------------: |
> | Linear |   61.71 ± 0.04   |   50.24 ± 0.45   |   67.34 ± 0.16   |   44.25 ± 0.13   |   85.02 ± 0.17   |
> | TIES   |   61.71 ± 0.05   |   50.00 ± 0.29   |   62.92 ± 0.09   |   47.76 ± 0.20   |   86.14 ± 0.13   |
> | Ours   | **64.10 ± 0.03** | **51.76 ± 0.09** | **67.97 ± 0.11** | **50.44 ± 0.09** | **86.23 ± 0.09** |
>
> The results show our method still achieves better performance compared with baseline methods.

---

> > ### Comment · Reviewer_aFmm · 2024-11-21
> >
> > Thanks for your reply. Regarding A3, it seems that your reference to A-OKVQA is incorrect.
> >
> > In addition, the questions in Q4 are actually based on the claims of ScienceQA and M3CoT. A large part of ScienceQA only contains pictures or only text. In addition, Chen et al. [1] also pointed out that more than 50% of ScienceQA's data can provide answers without relying on pictures. You may have to remove this part of the data to ensure that your conclusion is true and reliable.
> >
> > Additionally, you may need to strengthen some discussion of Q1 and Q2 in related work. I have no other questions that may need to be worried.
> >
> > I have modified my rating.
> >
> > [1] Chen et al. M3CoT: A Novel Benchmark for Multi-Domain Multi-step Multi-modal Chain-of-Thought. In ACL 2024.

---

> > > ### Author Response · Authors · 2024-11-21
> > >
> > > Thank you very much for raising your score. This is encouraging for us. If you have any further questions, please let us know. We will also be sure to include the suggested discussion in the revised version.

---

### Official Review · Reviewer_ck8W · 2024-11-04

**Soundness:** 2
**Presentation:** 3
**Contribution:** 2
**Rating:** 5
**Confidence:** 4

**Summary:**

The paper proposes an instance-specific way to merge vision-language models during inference. They rank models based on uncertainties and iteratively merge models in order. Results demonstrate the effectiveness of their method compared with other merging baselines.

**Strengths:**

1. The idea of having an instance-specific merging method is intuitive and can potentially inspire other methods in a similar direction.
2. They conduct a good number of experiments, with many baselines and tasks evaluated.
3. The writing is clear. For example, the experimental sections clearly state the research questions and answers for each part.

**Weaknesses:**

1. The method appears time-intensive due to 1) its instance-specific nature, requiring model merging score computation for each input; 2) uncertainty scores that rely on multiple perturbations per step; and 3) an iterative merging process. Further discussion on time complexity and inference computations is necessary.
2. The approach seems heuristic, and its effectiveness isn’t immediately clear. For instance, while the authors note that current merging methods may include over-confident, potentially harmful models, it's unclear how their method mitigates this. Quantitative definitions and results supporting their hypothesis would strengthen their case.
3. The experimental setup raises concerns. They split the LLaVA instruction tuning datasets into groups and train models on these splits for merging. However, their best-merged model significantly underperforms LLaVA-1.5 on some tasks (e.g., VQAv2: 71.77 vs. 78.5). Additionally, some results appear within the average ± standard deviation range of baseline numbers, without clear improvements over baselines.

**Questions:**

1. Does your method assume that some models are poorly calibrated, and would it still be effective if all models were well-calibrated?
2. What are the oracle model merging numbers for each task?

---

> ### Author Response · Authors · 2024-11-19
> **Point-to-point Response to Reviewer ck8W Part 1**
>
> We appreciate Reviewer ck8W's feedback, especially finding our work inspiring and experiment comprehensive. We provide pointwise responses to address your questions:
>
> **Q1: The method appears time-intensive due to 1) its instance-specific nature, requiring model merging for each input; 2) uncertainty scores that rely on multiple perturbations per step; and 3) an iterative merging process.**
>
>
> **A1:** We respectfully disagree and would like to point out that **our method is not instance-specific**. As shown in Figure 2 and Algorithm 1, our method is 1) **model-specific** rather than instance-specific and 2) merges models **once** and applies the merged model to all the inputs. For UQ, our method is also efficient as it requires only a **one-time effort** to obtain the uncertainty of all candidates, which can **be amortized** after deployment, and has **no additional inference overhead**.
>
> **Q2: How to remove harmful models remains unclear.**
>
>
> **A2:** As described in Figure 2, Section 3.2, and Section 3.4, we exclude over-confident models by sorting models in descending order of uncertainty and pick the merged model with the lowest uncertainty. We have provided extensive and comprehensive quantitative definitions and results to show our performance improvements compared to existing state-of-the-art baselines.
>
> **Q3: Merged model underperforms LLaVA-1.5 on some tasks. Some results appear within the average ± standard deviation range of baseline numbers.**
>
> **A3:** This confusion could be addressed by clarifying our setting. Motivated by training single MLLMs from scratch can be very computation-heavy [1, 2], and direct merging may include harmful models [3], we aim to design a **training-free** merging method to aggregate knowledge from existing MLLMs selectively.
>
> To create MLLMs with different expertise, we used LLaVA-1.5 architecture and trained on tasks from different domains (Section 4.1 Model Preparation). Thus, the fair baseline methods we consider in experiments are 1) single trained model, and 2) baseline merging methods. We compare our method to the best single model and direct merging model in the table below, where our method outperforms these baselines.
>
> |                     |  Average  |   AI2D    | ScienceQA | SeedBench |  MMBench  |  OOD-CV   |
> | ------------------- | :-------: | :-------: | :-------: | :-------: | :-------: | :-------: |
> | Single Best         |   30.21   |   22.83   |   35.38   |   21.07   |   44.53   |   27.24   |
> | Merge w/o Selection |   56.90   |   50.55   |   67.39   |   59.64   |   69.98   |   36.93   |
> | Ours                | **58.05** | **51.75** | **68.07** | **60.56** | **70.35** | **39.52** |
>
> |                     |  Average  |   OKVQA   |  TextVQA  |    GQA    |   MMMU    |  VizWiz   |   VQAv2   |   POPE    |
> | ------------------- | :-------: | :-------: | :-------: | :-------: | :-------: | :-------: | :-------: | :-------: |
> | Single Best         |   46.97   |   27.05   |   23.15   | **61.73** |   29.22   |   44.25   |   60.03   |   83.39   |
> | Merge w/o Selection |   54.71   |   44.51   | **44.23** |   48.56   |   35.00   | **55.21** |   70.21   |   85.23   |
> | Ours                | **56.01** | **50.39** |   43.49   |   50.75   | **35.22** |   54.13   | **71.77** | **86.33** |
>
> For the second point, we argue that this is due to the downward fluctuation of performance, and the results of 3 runs for random model selections in Table 8 and Table 9 show that our method is consistently better compared to baselines.
>
> > [1] "Meta-Llama/Llama-3.2-11B-Vision-Instruct ⋅ Hugging Face", 2024. \
> > [2] Tong, S., et al, "Cambrian-1: A Fully Open, Vision-Centric Exploration of Multimodal LLMs," 2024. \
> > [3] Zhao, X., et al, “Model-GLUE: Democratized LLM Scaling for A Large Model Zoo in the Wild,” 2024.

---

> ### Author Response · Authors · 2024-11-19
> **Point-to-point Response to Reviewer ck8W Part 2**
>
> **Q4: Method applicability to well-calibrated models**
>
>
> **A4:** Thanks for raising this question. No, our method doesn't assume models are poorly calibrated and can still perform well when models are well-calibrated.
>
>
> | Tuning Datasets | OKVQA | OCRVQA |  GQA  | VQAv2 | **TextCaps** | A_OKVQA | **RefCOCO** | LLaVA-Instruct | **ShareGPT** |  VG   |
> | --------------- | :---: | :----: | :---: | :---: | :----------: | :-----: | :---------: | :------------: | :----------: | :---: |
> | Before Cal      | 0.138 | 0.042  | 0.230 | 0.060 |  **0.659**   |  0.239  |  **0.106**  |     0.312      |  **0.101**   | 0.098 |
> | After Cal       | 0.113 | 0.041  | 0.203 | 0.058 |  **0.422**   |  0.223  |  **0.098**  |     0.282      |  **0.099**   | 0.096 |
>
>
> |            | Average | AI2D  | ScienceQA | OKVQA | POPE  |
> | ---------- | :-----: | :---: | :-------: | :---: | :---: |
> | Before Cal |  64.14  | 51.75 |   68.07   | 50.39 | 86.33 |
> | After Cal  |  63.86  | 51.12 |   67.71   | 50.44 | 86.18 |
>
>
> We tested the expected calibration error (ECE) of all models to merge before and after calibration [4], and **bold** means models excluded by our method. As shown in the results above: 1) the **ECE values of models to merge vary in big range**, showing that we are not only considering poorly calibrated models. 2) Our method excludes both a high ECE model and low ECE models, suggesting no inclination the calibration situation. 3) After calibration, our method selects the same group of models and shows **close performance before calibration** in the second table.
>
>
> > [4] Chuan Guo, et al, "On Calibration of Modern Neural Networks," 2017.
>
> **Q5: What are the oracle model merging numbers for each task?**
>
> **A5:** We would like to respectfully point out that this is infeasible. We have 10 models for merging and the oracle number requires 1023 runs for a single task, which takes **more than 1000 GPU hours** and is **unaffordable**  during the rebuttal session.

---

> ### Author Response · Authors · 2024-11-24
> **Genuine Request for Follow-up Discussion**
>
> Dear Reviewer ck8W,
>
> Thank you for all your efforts in reviewing our work. We have carefully addressed all your questions in detail. Please check our response and let us know if you have any further questions.
>
> Best regards, \
> Authors

---

> ### Comment · Reviewer_ck8W · 2024-11-25
> **Clarifications**
>
> Thank you for the response! For point 1, I was referring to the fact that you need to compute $UQ$ for each instance. For further clarification, how is $\bar{UQ}$ computed? For Line 284-293, do you mean you average $UQ$ over all the test instances of these benchmarks and then merge them based on the score?
>
>
> For point 2, I understood how you filtered the uncertainty models, my point was not "how to remove harmful models remains unclear", but that you need to support the claim that there are indeed "over-confident, potentially harmful models" and your method does target removing them. In other words, you need to explain why getting a merged model with low uncertainty would remove "over-confident, potentially harmful models". If all the models are over-confident and wrong, shouldn't their merged model have low uncertainty and "harmful" behaviors regardless of how you merge them?
>
> For point 3, I understood your setting, but my point was that a baseline with weak performance (e.g. ~70 VQAv2 score) is unreliable.

---

> > ### Author Response · Authors · 2024-11-25
> >
> > Thank you for your response. We provide our pointwise response as below:
> >
> > **For point 1:** To calculate the  $\bar{UQ}$ for a model, we compute the model‘s uncertainty on each sample from UQ datasets following Equation 1 on Line 211 to Line 215, and then we average the model’s uncertainty on all the samples from all the datasets as the $\bar{UQ}$. This $\bar{UQ}$ is viewed as the uncertainty for a model. We do not require merging for each input sample. And in the reply to [Reviewer 8odd](https://openreview.net/forum?id=SO0manOwUF&noteId=ygOiSaVNuP) we further showed our method can use up to only 5% samples of the UQ datasets, demonstrating a good data efficiency.
> >
> > **For point 2:** Stop merging with a low uncertainty could exclude over-confident models because: 1) the merging process starts from the single model with the highest uncertainty, and the uncertainty decreases by aggregating more knowledge from merging. 2) When over-confident models are merged, the performance of the merged model decreases (Table 2), leading to larger uncertainty. Your extreme example indeed challenges our algorithm, however, we argue that in real-world applications fine-tuned models will not all be over-confident and wrong and our method aims to filter out these models.
> >
> > **For point 3:** As we aim to propose a new merging method when combining datasets for training is infeasible, we compared our method with merging methods as baselines. Further, the baseline performance (70.21) already surpasses the largest BLIP-2 (65.2) [1], a model introduced in 2023, showing that the baseline performance is not unreliably weak.
> >
> > > [1] Junnan Li et al, "BLIP-2: Bootstrapping Language-Image Pre-training with Frozen Image Encoders and Large Language Models," 2023.

---

> ### Comment · Reviewer_ck8W · 2024-11-25
>
> > we average the model’s uncertainty on all the samples from all the datasets
>
> do these samples include the test instances?
>
> > Your extreme example indeed challenges our algorithm
>
> My point was exactly that there are cases where your algorithm wouldn't work as you claimed, thus it felt unconvincing and not technically sound to me.
>
> My previous question "Does your method assume that some models are poorly calibrated" was exactly trying to figure out what assumptions you need to make for your methods to work.
>
> > baseline performance (70.21) already surpasses the largest BLIP-2 (65.2)
>
> The largest BLIP-2 achieves 65.2 VQAv2 score in their zero-shot setting and ~82 VQAv2 score after finetuning. Your training dataset includes VQAv2 and should not be compared with their zero-shot number.
>
>
> > We have 10 models for merging and the oracle number requires 1023 runs for a single task,
>
> You can do a greedy search as you did for your algorithm.

---

> > ### Author Response · Authors · 2024-11-25
> >
> > Thank you for your response. We provide our pointwise response below:
> >
> > **Point 1:** Yes, these samples include test instances. But as our method doesn’t require labels for UQ, it will not cause data leakage.
> >
> > **Point 2:** As we show in [A4](https://openreview.net/forum?id=SO0manOwUF&noteId=E3M4v1zCuC), our method does not rely on assumptions of calibration situations for models. Our point is that 1) in real-world applications finetuned models are very unlikely to be all wrongly trained, and 2) when not all models are wrong, our method can still work to exclude these harmful models.
> >
> > **Point 3:** Sorry for the confusion. But our point is that the performance of baselines and our method are not unreasonably low. Further, we tested the official LLaVA-v1.5-7B on datasets not used in training, and the results are below:
> >
> > |                         | AI2D     | ScienceQA | SeedBench | TextVQA  | MMMU     | VizWiz   | POPE     |
> > | ----------------------- | -------- | --------- | --------- | -------- | -------- | -------- | -------- |
> > | LLaVA-V1.5, official    | **54.8** | **70.4**  | 60.5      | **46.1** | **35.3** | 54.4     | 85.9     |
> > | Single Best             | 22.8     | 35.4      | 21.1      | 23.2     | 29.2     | 44.3     | 83.4     |
> > | w/o selection, baseline | 50.6     | 67.4      | 59.6      | 44.2     | 35.0     | **55.2** | 85.2     |
> > | Ours                    | 51.8     | 68.1      | **60.6**  | 43.5     | 35.2     | 54.1     | **86.3** |
> >
> > From the results above, the performance of official LLaVA, baseline that doesn’t select, and our method achieve close performance on these benchmarks, showing that we are not comparing with unreasonably weak performance.
> >
> > On datasets used for training, the performance gap is within our expectation, because training on the combined dataset will bring mutual performance gain. However, our method aims to solve the challenge when combining datasets is infeasible.
> >
> > **Point 4:** Our algorithm does not do greedy search. Greedy search means trying all model combinations, which requires $\sum_{k=1}^n\binom{n}{k}=1023$ steps for $n=10$ models. But our algorithm reduces the search to 20 steps for $n=10$ models.

---

> ### Author Response · Authors · 2024-11-29
> **Thank You & Looking forward to Further Discussion**
>
> Dear Reviewer ck8W,
>
> We appreciate your insightful review and dedication to the review process. Due to the approaching deadline for rebuttal, we would like to respectfully convey our willingness to address any further concerns you may have. In our previous response, we carefully studied your comments and made detailed responses summarized below:
>
> - We used test instances but labels are not used.
> - We have no assumption of calibration. And in real-world applications where not all models are wrongly trained, our method could exclude harmful models.
> - Close performance with LLaVA-v1.5 on datasets not used for training.
> - We don’t rely greedy search in our algorithm.
>
> Thank you once again for your support.
>
> Warm regards, \
> Authors

---

> ### Author Response · Authors · 2024-11-30
> **Looking forward to Your Reply!**
>
> Dear Reviewer ck8W,
>
> We are very grateful for your constructive comments. We have made a substantial effort in responding to your questions and listed all the paper revisions and additional experiments in the [general response](https://openreview.net/forum?id=SO0manOwUF&noteId=AVlLH4kHW8). As the deadline for rebuttal is approaching, we sincerely hope you can provide us feedback before the discussion phase ends, and we are happy to answer any follow-up questions.
>
> Best regards, \
> Authors

---

> ### Author Response · Authors · 2024-12-02
>
> Dear Reviewer ck8W,
>
> We sincerely thank you for your valuable comments and time invested in our work. We have revised our paper and added relevant discussions and experiments. At present, all your concerns are responded to in the rebuttal and revised version of the paper. However, as there is **only 1 day remaining**, we kindly request your feedback to confirm that our response and revision effectively address your concerns. If there are any remaining issues, we would greatly appreciate the opportunity to address them to ensure the quality of our work. **We sincerely hope that you find our response convincing. Please consider revisiting your rating**.
>
> Best regards, \
> Authors

---

### Official Review · Reviewer_V122 · 2024-11-05

**Soundness:** 3
**Presentation:** 3
**Contribution:** 3
**Rating:** 5
**Confidence:** 3

**Summary:**

The paper  presents a novel approach for merging Multimodal Large Language Models (MLLMs). The proposed UQ-Merge algorithm addresses the limitations of existing model merging techniques, which often lead to suboptimal performance in vision-language tasks due to the inclusion of overconfident, less reliable models. UQ-Merge incorporates uncertainty quantification (UQ) to identify beneficial candidates, determine merging order, and optimize merging by considering both text and vision inputs. The method uses perturbation-based uncertainty assessments, allowing for more practical applications without the need for labeled data. Experimental results show that UQ-Merge significantly improves model accuracy across multiple benchmarks, outperforming traditional single-modal merging techniques.

**Strengths:**

1. The concept of using uncertainty quantification (UQ) to guide the merging of multimodal large language models (MLLMs) is relatively novel.
2. The paper is generally well-organized, with clear figures illustrating the architecture and merging process, as well as tables comparing the performance of different methods.

**Weaknesses:**

1. The paper does not provide a baseline performance for a single model trained on the combined dataset. If a single model achieves satisfactory results, there would be no need to follow the authors' approach of fine-tuning multiple models on different datasets within the same model architecture and then merging them.
Experiments were only conducted on LLaVA-1.5, without considering other models such as Intern-VL, Qwen2-VL, and MiniCPM-V, which reduces the persuasiveness of the results.
The authors’ merging method is limited to models obtained by fine-tuning the same model architecture on different datasets, without exploring merges between different model architectures, such as combinations of Intern-VL, Qwen2-VL, and LLaVA-1.5. The applicability of the proposed uncertainty quantification across different model families is also not considered.

**Questions:**

Please check the weakness.

---

> ### Author Response · Authors · 2024-11-19
> **Point-to-point Response to Reviewer V122**
>
> Thank you Reviewer V122 for acknowledging our work as novel and well-organized. To address Reviewer V122’s questions, we provide pointwise responses below:
>
> **Q1 Missing baseline of single model trained on combined dataset**
>
> **A1:** In this study, our goal is to develop a **training-free method** for aggregating knowledge from pre-trained models created by other users, as outlined in our Abstract and Introduction. Training on combined datasets can be prohibitively expensive [1, 2], and current model merging methods often include over-confident models, as shown in Figure 1 and Table 2. To address this, we propose a training-free UQ-based approach that **efficiently aggregates knowledge** while **excluding harmful models**.
>
> > [1] "Meta-Llama/Llama-3.2-11B-Vision-Instruct ⋅ Hugging Face", 2024. \
> > [2] Tong, S., et al, "Cambrian-1: A Fully Open, Vision-Centric Exploration of Multimodal LLMs," 2024.
>
> **Q2 Experiment on more model families and applicability of UQ to other models.**
>
> **A2:** Thank you for the insightful comment. We fine-tuned InternVL-2-1B using the same datasets described in Section 4.1 (Model Preparation) and applied our method to the fine-tuned models. The results are shown below:
>
> | Methods |  Average  |   AI2D    | ScienceQA |   OKVQA   |   POPE    |
> | ------- | :-------: | :-------: | :-------: | :-------: | :-------: |
> | Linear  |   68.25   |   60.56   |  **84.88**   |   39.75   |   87.82   |
> | TIES    |   59.69   |   49.03   |   73.77   |   28.55   |   87.42   |
> | Ours    | **68.60** | **61.24** | **84.88** | **40.19** | **88.07** |
>
> These results demonstrate that our perturbation-based UQ **can generalize to other model family**, both shown in the result above and in Section 3.3 that input perturbation and model perturbation don't rely on specific architectures.
>
> **Q3: Exploring merging different model architectures.**
>
> **A3:** Thanks for raising a good question. In the MLLM domain, model merging is still under-explored and a novel area, as acknowledged by [**Reviewer 8odd**](https://openreview.net/forum?id=SO0manOwUF&noteId=ygOiSaVNuP) and [**Reviewer aFmm**](https://openreview.net/forum?id=SO0manOwUF&noteId=Fydbq0mUjA). Our study addresses the two fundamental challenges in model merging: 1) candidate selection and 2) pipeline optimization (including merging method and process). While merging different architectures presents an intriguing area for future research, current merging techniques primarily operate on models with identical architectures [3]. We plan to explore MoE-based approaches for merging different architectures in future research [4, 5].
>
> > [3] Goddard, C., et al. “Arcee’s MergeKit: A Toolkit for Merging Large Language Models,” 2024. \
> > [4] Ding, N., et al, "Mastering Text, Code and Math Simultaneously via Fusing Highly Specialized Language Models," 2024. \
> > [5] Sukhbaatar, S., et al, "Branch-Train-MiX: Mixing Expert LLMs into a Mixture-of-Experts LLM," 2024.

---

> > ### Comment · Reviewer_V122 · 2024-11-26
> > **Thanks for your reply, I would maintain my rating.**
> >
> > 1. "Training-free" does not convince me. In practice, it is very strange that train models with same architecture using different datasets, but ignore the strategy that directly train a single model with all of relevant datasets. This is the most important baseline but missing in the paper.

---

> > > ### Author Response · Authors · 2024-11-26
> > >
> > > Thank you for your response. We provide our reply from two points:
> > >
> > > **Point 1: Train on different datasets rather than on combined datasets.** We would like to point out that this is a common and practical practice in the real world.
> > >
> > > In the LLM domain, there exist numerous LLMs finetuned on different datasets from the same architecture. For example, starting from the pretrained Llama 2 model, there are:
> > >
> > > | Model Name                                   | Data Availability |
> > > | -------------------------------------------- | :---------------: |
> > > | migtissera/Synthia-7B-v1.2                   |        No         |
> > > | neuralmagic/Llama-2-7b-evolcodealpaca        |        Yes        |
> > > | teknium/OpenHermes-7B                        |        Yes        |
> > > | PygmalionAI/pygmalion-2-7b                   |        Yes        |
> > > | meta-llama/Llama-2-7b-chat-hf                |        No         |
> > > | Severus27/BeingWell_llama2_7b                |        Yes        |
> > > | meta-math/MetaMath-7B-V1.0                   |        Yes        |
> > > | lmsys/vicuna-7b-v1.5                         |        No         |
> > > | garage-bAInd/Platypus2-7B                    |        Yes        |
> > > | GOAT-AI/GOAT-7B-Community                    |        No         |
> > > | stanford-oval/Llama-2-7b-WikiChat-fused      |        No         |
> > > | cognitivecomputations/dolphin-llama2-7b      |        Yes        |
> > > | meta-math/MetaMath-Llemma-7B                 |        Yes        |
> > > | codellama/CodeLlama-7b-Instruct-hf           |        No         |
> > > | ise-uiuc/Magicoder-S-CL-7B                   |        Yes        |
> > > | LLM360/CrystalChat                           |        Yes        |
> > > | synapsoft/Llama-2-7b-hf-flan2022-1.2M        |        No         |
> > > | GenSEC-LLM/SLT-Task1-Llama2-7b-HyPo-baseline |        No         |
> > > | TheBloke/Llama-2-7B-GGML                     |        No         |
> > > | allenai/tulu-2-dpo-7b                        |        Yes        |
> > >
> > > However as shown in the table above, not all the datasets are available, hindering the practice of combining datasets and training.
> > >
> > > Another example is LoRA Hub [1], which aims to leverage existing LoRA modules trained on diverse given tasks from third-party users. We share a similar aim of incorporating models from third parties where the availability of datasets is not guaranteed.
> > >
> > > > [1] Chengsong Huang et al, "LoraHub: Efficient Cross-Task Generalization via Dynamic LoRA Composition," 2024.
> > >
> > > Similar to [2, 3, 4], our goal is to **leverage existing Multimodal LLMs** in a training-free way, when **combining datasets is infeasible** or the **training cost is too high**.
> > >
> > > > [2] Mitchell Wortsman et al, "Model soups: averaging weights of multiple fine-tuned models improves accuracy without increasing inference time," 2022. \
> > > > [3] Prateek Yadav et al, "TIES-Merging: Resolving Interference When Merging Models," 2023. \
> > > > [4] Gabriel Ilharco et al, "Editing Models with Task Arithmetic," 2023.
> > >
> > > **Point 2: Comparison with a baseline trained on a combined dataset.** Thank you for your suggestion. We were running this experiment previously and now we have got the test result on unseen datasets (not used for training) and will include this in our revised paper.
> > >
> > > |                  | AI2D     | ScienceQA | SeedBench | TextVQA  | MMMU     | VizWiz   | POPE     |
> > > | ---------------- | -------- | --------- | --------- | -------- | -------- | -------- | -------- |
> > > | Combined dataset | **54.8** | **70.4**  | 60.5      | **46.1** | **35.3** | **54.4** | 85.9     |
> > > | w/o selection    | 50.6     | 67.4      | 59.6      | 44.2     | 35.0     | 55.2     | 85.2     |
> > > | Ours             | 51.8     | 68.1      | **60.6**  | 43.5     | 35.2     | 54.1     | **86.3** |
> > >
> > > From the results above we can see that our method 1) performs better compared to merging w/o selection, and 2) achieves close or surpassing performance compared to baseline trained on the combined dataset, suggesting the effectiveness of our method to aggregate knowledge in a training-free way.

---

> ### Author Response · Authors · 2024-11-24
> **Genuine Request for Follow-up Discussion**
>
> Dear Reviewer V122,
>
> We appreciate your efforts in reviewing our paper. We have tried our best to address all your questions in detail. Please check our response and let us know if you have further questions. Once again, thank you very much for your time, help, and consideration.
>
> Best regards, \
> Authors

---

> ### Author Response · Authors · 2024-11-29
> **Thank You & Looking forward to Further Discussion**
>
> Dear Reviewer V122,
>
> We appreciate your insightful review and dedication to the review process. Due to the approaching deadline for rebuttal, we would like to respectfully convey our willingness to address any further concerns you may have. In our previous response, we carefully studied your comments and made detailed responses summarized below:
>
> - Explanation of model merging and examples that combining datasets and train is infeasible.
> - Comparison with a baseline trained on a combined dataset.
>
> Thank you once again for your support.
>
> Warm regards, \
> Authors

---

> ### Author Response · Authors · 2024-11-30
> **Looking forward to Your Reply!**
>
> Dear Reviewer V122,
>
> We are very grateful for your constructive comments. We have made a substantial effort in responding to your questions and listed all the paper revisions and additional experiments in the [general response](https://openreview.net/forum?id=SO0manOwUF&noteId=AVlLH4kHW8). As the deadline for rebuttal is approaching, we sincerely hope you can provide us feedback before the discussion phase ends, and we are happy to answer any follow-up questions.
>
> Best regards, \
> Authors

---

> ### Author Response · Authors · 2024-12-02
>
> Dear Reviewer V122,
>
> We sincerely thank you for your valuable comments and time invested in our work. We have revised our paper and added relevant discussions and experiments. At present, all your concerns are responded to in the rebuttal and revised version of the paper. However, as there is **only 1 day remaining**, we kindly request your feedback to confirm that our response and revision effectively address your concerns. If there are any remaining issues, we would greatly appreciate the opportunity to address them to ensure the quality of our work. **We sincerely hope that you find our response convincing. Please consider revisiting your rating**.
>
> Best regards, \
> Authors

---

### Author Response · Authors · 2024-11-27
**General Response**

We sincerely thank all the reviewers for their insightful comments and suggestions for improving our work. In addition to replying to specific reviewers, we would like to conclude our paper revision and rebuttal experiments so far.

**Paper Revision**

In the revised paper, we added contents below following reviewers’ suggestions, and new contents are marked in $\color{blue}{blue}$.

- Performance of the model trained on the combined dataset in Table 1 and Table 2. [[Reviewer V122](https://openreview.net/forum?id=SO0manOwUF&noteId=vlMPQYmmBX), [Reviewer ck8W](https://openreview.net/forum?id=SO0manOwUF&noteId=6Lu7XYIqtTJ)]
- Discussion about why exploring model merging for MLLMs and another UQ-related merging method in Related Work. [[Reviewer aFmm](https://openreview.net/forum?id=SO0manOwUF&noteId=Fydbq0mUjA)]

**Rebuttal Experiments**

In the rebuttal period, we have added more experiments to support the **effectiveness** and **data efficiency** of our method.

- Applying our method to InternVL2-1B. [[Reviewer V122](https://openreview.net/forum?id=SO0manOwUF&noteId=tVOON7F84J)]
- Comparison with a model trained on the combined dataset. [[Reviewer V122](https://openreview.net/forum?id=SO0manOwUF&noteId=q3cNIbTcvr), [Reviewer ck8W](https://openreview.net/forum?id=SO0manOwUF&noteId=8HVneS2CaO)]
- Effectiveness of our method before and after calibrating finetuned models. [[Reviewer ck8W](https://openreview.net/forum?id=SO0manOwUF&noteId=E3M4v1zCuC)]
- The performance w/ and w/o commonsense knowledge from A_OKVQA. [[Reviewer aFmm](https://openreview.net/forum?id=SO0manOwUF&noteId=mI0qXOZRbi)]
- Variation and error metrics across several runs. [[Reviewer aFmm](https://openreview.net/forum?id=SO0manOwUF&noteId=mI0qXOZRbi)]
- Replacing perturbation-based UQ with conformal prediction. [[Reviewer 8odd](https://openreview.net/forum?id=SO0manOwUF&noteId=ZDvgihlt7k)]
- Different ratios of UQ datasets. [[Reviewer 8odd](https://openreview.net/forum?id=SO0manOwUF&noteId=WCiiV9lVj7)]

---

### Note · Authors · 2024-12-14

I have read and agree with the venue's withdrawal policy on behalf of myself and my co-authors.